# The impact of stability considerations on genetic fine-mapping

Alan J Aw[1,2†], Lionel Chentian Jin[3], Nilah Ioannidis[2,4*], Yun S Song[1,2,4*]

[1]Department of Statistics, University of California, Berkeley, United States; [2]Center for Computational Biology, University of California, Berkeley, United States; [3]McKinsey & Company, New York, United States; [4]Computer Science Division, University of California, Berkeley, United States

## eLife Assessment

This **important** study presents a methodologically rigorous framework for stability-guided fine-mapping, extending PICS and generalizing to methods such as SuSiE, supported by comprehensive simulations and functional enrichment analyses. The evidence is now **convincing**, demonstrating improved causal variant recovery and offering a robust alternative for cross-population fine-mapping. The approach will be of particular interest to statistical geneticists, computational biologists, and biomedical researchers who rely on fine-mapping to interpret genetic association signals.

**\*For correspondence:**
nilah@berkeley.edu (NI);
yss@berkeley.edu (YSS)

**Present address:** [†]Department of Genetics, University of Pennsylvania, Philadelphia, United States

**Abstract** Fine-mapping methods, which aim to identify genetic variants responsible for complex traits following genetic association studies, typically assume that sufficient adjustments for confounding within the association study cohort have been made, for example, through regressing out the top principal components (i.e., residualization). Despite its widespread use, however, residualization may not completely remove all sources of confounding. Here, we propose a complementary stability-guided approach that does not rely on residualization, which identifies consistently fine-mapped variants across different genetic backgrounds or environments. Simulations show that stability guidance neither outperforms nor underperforms residualization, but each approach picks up different variants considerably often. Critically, prioritizing variants that match between the residualization and stability-guided approaches enhances recovery of causal variants. We further demonstrate the utility of the stability approach by applying it to fine-map eQTLs in the GEUVADIS data. Using 378 different functional annotations of the human genome, including recent deep learning-based annotations (e.g., Enformer), we compare enrichments of these annotations among variants for which the stability and traditional residualization-based fine-mapping approaches agree against those for which they disagree and find that the stability approach enhances the power of traditional fine-mapping methods in identifying variants with functional impact. Finally, in cases where the two approaches report distinct variants, our approach identifies variants comparably enriched for functional annotations. Our findings suggest that the stability principle, as a conceptually simple device, complements existing approaches to fine-mapping, reinforcing recent advocacy of evaluating cross-population and cross-environment portability of biological findings. To support visualization and interpretation of our results, we provide a Shiny app, accessible at https://github.com/songlab-cal/StableFM.

## Introduction

An important challenge faced by computational precision health research is the lack of generalizability of biological findings, which are often obtained by studying cohorts that do not include particular

communities of individuals. Known as the *cross-population generalizability* or *portability problem*, the challenge persists in multiple settings, including gene expression prediction (*Keys et al., 2020*) and polygenic risk prediction (*Mostafavi et al., 2020*). Generalizable biological signals, such as the functional impact of a variant, are important, as they ensure that general conclusions drawn from cohort-specific analyses are not based on spurious discoveries. Portability problems are potentially attributable to cohort-biased discoveries being treated as generalizable true signals. Efforts to address such problems have included the use of diverse cohorts typically representing multiple population ancestries (*Márquez-Luna et al., 2017*), meta-analyses of earlier studies across diverse cohorts (*Turley et al., 2021*; *Han and Eskin, 2012*; *Morris, 2011*; *Willer et al., 2010*), or focusing on biological markers that are more likely a priori to play a causal role (including proxy variables), for example, rare variants (*Zaidi and Mathieson, 2020*) or the transcriptome (*Liang et al., 2022*).

In this paper, we consider an approach based on the notion of *stability* to improve generalizability. Being a pillar of veridical data science (*Yu and Kumbier, 2020*), stability refers to the robustness of statistical conclusions to *perturbations* of the data. Perturbations are not arbitrary, but instead, they encode the practitioner's beliefs about the quality of the data and the nature of the relationship between variables. Well-chosen perturbation schemes will help the practitioner obtain statistical conclusions that are robust, in the sense that they are generalizable rather than spurious findings, and therefore more likely to capture the true signal. For example, in sparse linear models, prioritizing the stability of effect sizes to cross-validation 'perturbations' leads to a much smaller set of selected features without reducing predictive performance (*Lim and Yu, 2016*). In another example involving the application of random forests to detect higher-order interactions between gene regulation features (*Basu et al., 2018*), interactions stable to bootstrap perturbations are also largely consistent with known physical interactions between the associated transcription factor binding or enhancer sites.

To further investigate the utility of the stability approach, we focus on a procedure known as (genetic) fine-mapping (*Schaid et al., 2018*). Fine-mapping is the task of identifying genetic variants that causally affect some trait of interest. From the viewpoint of stability, previous works focusing on cross-population stability of fine-mapped variants typically use Bayesian linear models. The linear model has allowed modeling of heterogeneous effect sizes across different user-defined populations (e.g., ethnic groups, ancestrally distinct populations, or study cohorts), and it is common to assume that causal variants share correlated effects across populations (see, e.g., eq. (24) of *LaPierre et al., 2021* or eq. (9) of *Wen et al., 2015*).

Unlike the parametric approaches described above, we here apply a *non-parametric* fine-mapping method to GEUVADIS (*Lappalainen et al., 2013*), a database of gene expression traits measured across individuals of diverse genetic ancestries and from different geographical environments, whose accompanying genotypes are available through the 1000 Genomes Project (1 kGP) (*Auton et al., 2015*). We apply fine-mapping through two approaches, one that is commonly used in practice and another that is guided by stability. First, we perform simulations using 1kGP genotypes to evaluate the strength of each approach. We next evaluate the agreement of fine-mapped variants between the two approaches on GEUVADIS data, and then measure the functional significance of variants picked by both approaches using a much wider range of functional annotations than considered in previous studies. By performing various statistical tests on the functional annotations, we evaluate the advantages brought about by the incorporation of stability. Finally, we investigate the broader applicability of the stability approach by implementing and analyzing a stability-guided version of the fine-mapping algorithm SuSiE on simulated data. To support visualization of results obtained from the GEUVADIS analysis at both the single gene and genome-wide levels, we provide an interactive Shiny app which is accessible at: https://github.com/songlab-cal/StableFM. Our app is open-source and designed to support geneticists in interpreting our results, thereby also addressing a growing demand for accessible, *integrative* software to interpret genetic findings in the age of big data and variant annotation databases.

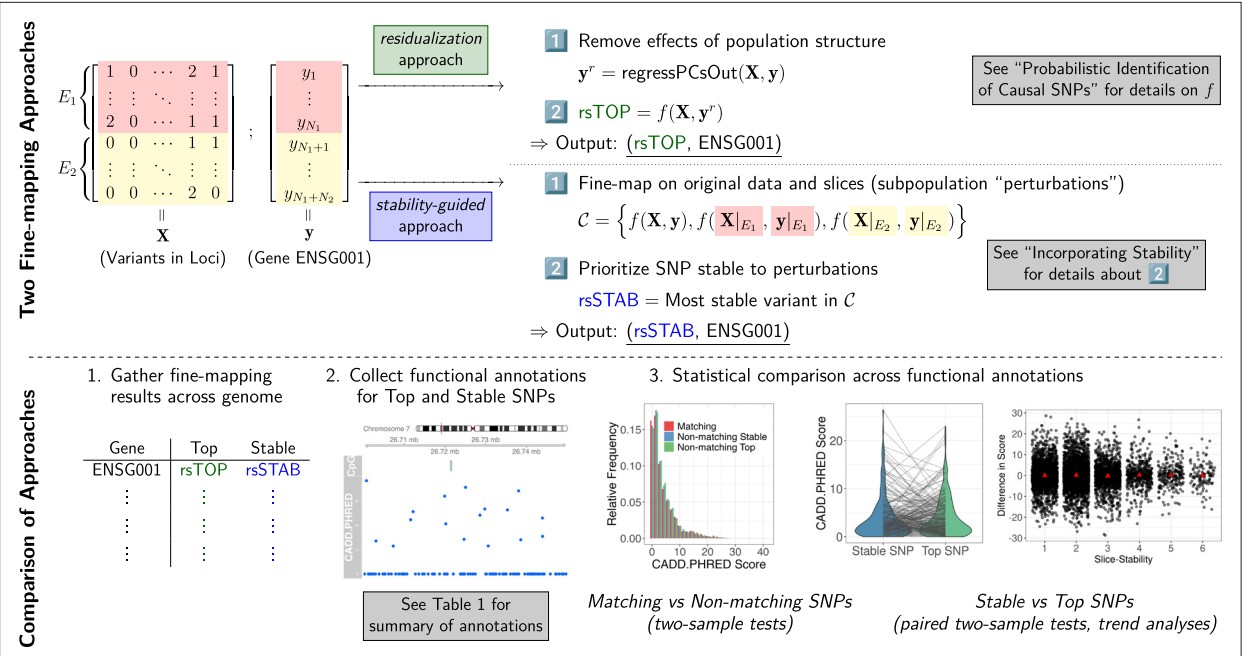

**Figure 1.** An overview of our study of the impact of stability considerations on genetic fine-mapping. (**A**) The two ways in which we perform fine-mapping, the first of which (colored in green) prioritizes the stability of variant discoveries to subpopulation perturbations. The data illustrates the case where there are two distinct environments, or subpopulations (denoted $E_1$ and $E_2$), that split the observations. (**B**) Key steps in our comparison of the stability-guided approach with the popular residualization approach.

## Results

### Experimental design

We apply the fine-mapping method PICS (*Taylor et al., 2021*; *Farh et al., 2015*; see Algorithm 1 in Probabilistic Identification of Causal SNPs) to the GEUVADIS data (*Lappalainen et al., 2013*), which consists of $T = 22,720$ gene expression traits measured across $N = 445$ individuals, with their accompanying genotypes obtained from the 1000 Genomes Project (*Auton et al., 2015*). These individuals are of either European or African ancestry, with about four-fifths of the cohort made up of individuals of (self-identified) European descent. In particular, these ancestrally different subpopulations have distinct linkage disequilibrium patterns and environmental exposures, which constitute potential confounders that we wish to stabilize the fine-mapping procedure against.

PICS, like many eQTL analysis methods, requires the lead variant at a locus to compute posterior probabilities, so we perform marginal regressions of each gene expression trait against variants within the fine-mapping locus. Our implementation of PICS generates three sets of variants, which represent putatively causal variants that are (marginally) highly associated, moderately associated, and weakly associated with the gene expression phenotype. (See Materials and methods for details.) As illustrated in *Figure 1A*, we compare two versions of the fine-mapping procedure—one that is typically performed in practice, and another that is motivated by the stability approach.

### Stable variant

Specifically, to incorporate stability into fine-mapping and obtain what we call the *stable variant* or stable SNP, we encode into the algorithm our belief that for a gene expression trait, a causal variant would presumably act through the same mechanism regardless of the population from which they originate. Indeed, if a set of genetic variants were causal, the same algorithm should report it, if run on subsets of the data corresponding to heterogeneous populations. This belief is consistent with recent simulation studies showing that the inclusion of GWAS variants discovered in diverse populations both mitigates false discoveries driven by linkage disequilibrium differences (*Li et al., 2022*) and improves generalizability of polygenic score construction (*Cavazos and Witte, 2021*). Hence, the stable variant is the variant with the highest probability of being causal, conditioned on being reported by PICS in

the most number of subpopulations. Incorporating stability provides a formal definition of the stable variant.

## Top variant

The stability consideration we have described is closely related to a popular procedure known as correction for population structure, which residualizes the trait using measured confounders (e.g., top principal components computed from the genotype matrix). We call the variant returned by the residualization approach the *top variant* or top SNP. The residualization approach removes the effects of genetic ancestry and environmental exposures on a trait and is used to avoid the risk of low statistical power borne by stratified analyses. The top variant is formally defined toward the end of PICS. We remark that the top variant is a function of the residualized trait, as opposed to the stable variant, which is a function of the unresidualized trait (see *Figure 1A*).

## Simulation study

We simulated 100 genes from GEUVADIS gene expression data, sampled proportionally across the 22 autosomes. Mirroring the approach described in Section 4 of *Wang et al., 2020*, synthetic gene expression phenotypes were simulated based on the sampled genes and involved two parameters: $C$, the number of causal variants, and $\phi$, the proportion of variance in gene expression explained by variants in cis. Whenever available, gene expression canonical TSS was used to include only variants lying within 1 Mb upstream and downstream when simulating gene expression phenotypes. We considered all combinations of $C \in \{1, 2, 3\}$ and $\phi \in \{0.05, 0.1, 0.2, 0.4\}$, and simulated two replicates for each gene and for each combination of $C$ and $\phi$. Altogether, we generated $100 \times 3 \times 4 \times 2 = 2400$ datasets, on which we ran PICS and SuSiE. For PICS, we ran four versions: the *stability-guided* version and *residualization* version, which return the stable variant and top variant, respectively; a *combined* version, which runs stability-guided PICS on covariate-residualized phenotypes; as well as a *plain* version, where no correction for population stratification or stability guidance was applied. For SuSiE, we ran two versions: the stability-guided version and the residualization version. We also ran a separate set

**Table 1.** A list of 378 functional annotations across which the biological significances of stable and top fine-mapped single nucleotide polymorphisms are compared.

Annotations that report multiple scores have the total number of scores reported shown in parentheses. Scores mined from the FAVOR database (*Zhou et al., 2023*) are indicated by an asterisk (TSS = transcription start site, bp = base pair).

| Functional annotation type | Functional annotation |
|---|---|
| Ensembl | Distance to Canonical *TSS* (*Cunningham et al., 2022*) |
| | Regulatory Features (6; *Cunningham et al., 2022*) |
| Computational predictions | CADD* (2; *Rentzsch et al., 2019*) |
| | SIFTVal* (*Ng and Henikoff, 2003*) |
| | FATHMM-XF* (*Rogers et al., 2018*) |
| | LINSIGHT* (*Huang et al., 2017*) |
| | Polyphen* (*Adzhubei et al., 2010*) |
| | PhyloP* (3; *Pollard et al., 2010*) |
| | Gerp* (2; *Davydov et al., 2010*) |
| | B Statistic* (*McVicker et al., 2009*) |
| | FunSeq2* (*Fu et al., 2014*) |
| | ALoFT* (*Balasubramanian et al., 2017*) |
| | Percent CpG in 75 bp window* (*Rentzsch et al., 2019*) |
| | Percent GC in 75 bp window* (*Rentzsch et al., 2019*) |
| | FIRE (*Ioannidis et al., 2017*) |
| | Enformer (177 tracks × 2 scores per track; *Avsec et al., 2021*) |

of simulations comprising six scenarios of environmental heterogeneity by ancestry on 10 genes from Chromosomes 1, 20, 21, and 22 (6 × 10 × 3 × 4 × 2 = 1440 generated datasets), on which we ran the plain and stability-guided versions of PICS.

### Evaluation on simulated genes

Following *Mazumder, 2020*, we analyze the recovery probability of all fine-mapping algorithms as a function of the signal-to-noise ratio (SNR), defined here as the ratio of true signal to background noise: $\text{SNR} = \text{var}(\mathbf{Xb})/\sigma^2$ (see Simulation study and evaluation details in Materials and methods for details).

### Analysis of GEUVADIS data

We compare the results obtained by using each approach, across a range of categories of functional annotations including conservation, pathogenicity, chromatin accessibility, transcription factor binding, and histone modification, and across biological assays and computational predictions. See *Table 1* for a full list of annotations covered. (Appendix 5 contains a full description of each quantity and its interpretation.)

*Figure 1B* summarizes the key steps of our investigation. To be clear, a first comparison is between the set of genes for which the top and stable variants match and the set of genes for which the top and stable variants disagree; comparisons are made between matching variants and one of the non-matching sets of variants (top or stable). A second comparison is restricted to the set of genes with non-matching top and stable variants, that is, between *paired* sets of variants fine-mapped to a gene, where one set of a pair corresponds to the output of the residualization approach whereas the other corresponds to the output of the stability-guided approach. We additionally compare across three fine-mapping output sets, corresponding to variants that have a high, moderate, or weak marginal association with the expression phenotype. For brevity, we term these three sets Potential Set 1, Potential Set 2, and Potential Set 3, respectively. (See Algorithm 1 for details.) For each Potential Set, we find the top and stable variants as described in Incorporating stability.

## Simulations justify exploration of stability guidance

To better understand how stability guidance may support the discovery of causal variants across ancestrally diverse cohorts that possess confounding exogenous factors, we selected 10 genes from Chromosomes 1, 20, 21, and 22 to simulate six scenarios where environmental heterogeneity by ancestry influences gene expression. We consider four scenarios where environmental *variance* differed between ancestries and two scenarios where environmental *mean* differed between ancestries. We also vary the number of causal variants and proportion of variance in gene expression expressed in *cis*. We run Stable PICS, which returns stable variants; alongside a 'plain' version of PICS (Plain PICS), which neither performs PC-residualization of phenotypes nor incorporates stability.

We find that, across these simulations, performance was not significantly different between Stable PICS and Plain PICS: both approaches recover the true causal variant in Potential Set 1 with similar frequencies in simulations involving one causal variant, and in simulations involving two or more causal variants the leading potential sets recover at least one causal variant with similar frequency (*Figure 2A*). These also hold for lower potential sets (*Figure 2—figure supplement 1*). While the two approaches agree more often than not, there is greater disagreement when considering simulations with low SNR (which may potentially reflect real gene expression data—see Stable variants frequently do not match top variants in GEUVADIS). For example, we identified 83% matching variants in Potential Set 1 across all simulations, but this dropped to 74% when restricting to simulations with $\phi = 0.05$. We next split results by whether the variant returned by Plain PICS had a large posterior probability (PP)—defined as >0.9 to reflect thresholds reported in practice—and found strong agreement between the two approaches when PP was large: for Potential Set 1 the agreement was 98%, Potential Set 2 was 90%, and Potential Set 3 was 93% (*Appendix 12—table 1*). These observations together imply that the two approaches disagree particularly when SNR is low and the PP of variants returned by Plain PICS is not large, and Stable PICS recovers causal variants accurately in certain cases where Plain PICS fails to do so (otherwise their performances would be different).

Looking closely at each simulation scenario, we see that Stable PICS recovers a causal variant more frequently than Plain PICS in scenarios where there are multiple causal variants and environmental heterogeneity is driven by a 'spiked mean shift' (Simulation study and evaluation details),

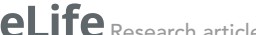

**Figure 2.** Simulation study results. (**A**) The frequency with which at least one causal variant is recovered in Potential Set 1 by Plain PICS and Stable PICS, across 1440 simulated gene expression data that incorporate ancestry-mediated environmental heterogeneity. Recovery frequencies are stratified by simulations differing in the number of causal variants, and the Venn diagram reports the number of matching and non-matching variants in Potential Set 1 across all simulations. (**B**) The frequency with which at least one causal variant is recovered in Potential Set 1 by Combined PICS, Stable PICS, and Top PICS, across 2400 simulated gene expression data. Recovery frequencies are stratified by the SNR parameter $\phi$ used in simulations, and the Venn diagram reports the number of matching and non-matching variants in Potential Set 1 across all simulations. (**C**) The frequency with which at least one causal variant is recovered in Credible Set 1 by Stable SuSiE and Top SuSiE. Venn diagram reports the number of matching and non-matching variants in Potential Set 1 across all simulations. (**D**) The frequency with which matching and non-matching variants in the first credible or potential set recover a causal variant, obtained from comparing top and stable approaches to an algorithm. We report approximate 95% confidence intervals for each point estimate, by multiplying the associated standard error of the estimate by 1.96.

The online version of this article includes the following figure supplement(s) for figure 2:

**Figure supplement 1.** Plain PICS vs Stable PICS (Potential Sets 2 and 3).

**Figure supplement 2.** Performance of PICS algorithms (Potential Sets 2 and 3).

*Figure 2 continued on next page*

**Figure supplement 3.** Performance of SuSiE algorithms (Credible Sets 2 and 3).

**Figure supplement 4.** Matching vs non-matching variants (Potential and Credible Sets 1 and 2).

**Figure supplement 5.** Stable PICS vs Stable SuSiE (one causal variant).

**Figure supplement 6.** Stable PICS vs Stable SuSiE (two causal variants).

**Figure supplement 7.** Stable PICS vs Stable SuSiE (three causal variants).

**Figure supplement 9.** Non-matching Top vs Stable SNP posterior probabilities.

**Figure supplement 10.** Stable PICS vs SuSiE (one causal variant).

**Figure supplement 11.** Stable PICS vs SuSiE (two causal variants).

**Figure supplement 12.** Stable PICS vs SuSiE (three causal variants).

**Figure supplement 19.** Variant recovery frequency of PICS and SuSiE matching and non-matching variants (one causal variant).

**Figure supplement 31.** Plain PICS vs Stable PICS in 'mean shift ($|i - 3|$)' environmental heterogeneity simulations.

while Plain PICS generally outperforms when environmental heterogeneity is driven by differences in environmental variation. For example, in a set of spiked mean shift simulations with three causal variants, with low SNR ($\phi = 0.1$) and with exogenous variable drawn from $N(2\sigma, \sigma^2)$ for only GBR individuals (rest are drawn from $N(0, \sigma^2)$), the frequencies at which at least one causal variant is recovered were 80% and 75% for Stable PICS and Plain PICS, respectively (see *Figure 2—figure supplement 32*). However, none of these differences in performance is statistically significant. In summary, these findings demonstrate that non-genetic confounding in cohorts can reduce power in methods not adjusting or accounting for ancestral confounding but can be remedied by approaches that do so. (Performances stratified by SNR and number of potential sets included are summarized in *Figure 2—figure supplements 24–26*; performances stratified by each scenario are summarized in *Figure 2—figure supplements 27–32*).

## Simulations reveal advantages (and disadvantages) of stability guidance

We next conduct a larger set of simulations using 100 genes selected across all 22 autosomes. Similar to the earlier set of simulations, we vary both the number of causal variants and proportion of variance in gene expression explained by variants in *cis*. However, here we compare stability-guided fine-mapping against residualization fine-mapping, a commonly used approach.

## No difference in power between stability guidance and residualization, although considerably many variants do not match

Comparing stable and top variants returned by Stable PICS and PICS run via the residualization approach (Top PICS), we observed similar causal variant recovery rates. Across all 2400 simulated gene expression phenotypes, the stable variant in Potential Set 1 was causal with frequency 0.63, which is close to and not significantly different from the frequency at which the Potential Set 1 top variant was causal (0.64; McNemar test p-value = 0.38). Similarly close frequencies were observed for lower potential sets (Potential Set 2: stable = 0.12, top = 0.13; Potential Set 3: stable = 0.041, top = 0.047), as well as when results were stratified by SNR (*Figure 2B*). However, top and stable variants often disagreed, with Potential Set 1 having 72% matching top and stable variants, Potential Set 2 matching at 40% and Potential Set 3 matching at 25% (*Figure 2—figure supplement 2*). Stratifying results by simulation parameters (number of causal variants, $S$, and SNR, $\phi$), we generally observed matching fractions increasing with SNR, at least in Potential Set 1: for example, in simulations involving one causal variant, the matching fraction is 69.5% under $\text{SNR} = 0.05/(1 − 0.05) ≈ 0.053$, and it increases monotonically to 89.5% under $\text{SNR} = 0.4/(1 − 0.4) ≈ 0.67$ (*Appendix 12—table 2*). There was no clear relationship between matching fractions and the number of causal variants simulated.

## Searching for matching variants between Top PICS and Stable PICS improves causal variant recovery

Given that stable and top variants frequently do not match despite achieving similar performance, we suspect that, similar to the smaller set of simulations on which Plain PICS and Stable PICS were compared, each approach can recover causal variants in scenarios where the other does not. We thus explore ways to combine the residualization and stability-driven approaches, by considering (1) combining them into a single fine-mapping algorithm (we call the resulting procedure *Combined PICS*); and (2) prioritizing matching variants between the two algorithms. Comparing the performance of Combined PICS against both Top and Stable PICS, however, we find no significant difference in its ability to recover causal variants (*Figure 2B*). This conclusion held even when we analyzed performance by stratifying simulations by SNR and number of causal variants simulated ($\phi$ and $S$ parameters; see *Figure 2—figure supplements 13–15*). On the other hand, matching variants between Top and Stable PICS are significantly more likely to be causal. Across all simulations, a matching variant in Potential Set 1 was 2.5× as likely to be causal than either a non-matching top or stable variant (*Figure 2D*)—a result that was qualitatively consistent even when we stratified simulations by SNR and number of causal variants simulated (*Figure 2—figure supplements 19, 20* and *Figure 2—figure supplement 22*). A similar trend was observed for Potential Set 2, although we do not see much higher causal variant recovery when comparing matching and non-matching variants in Potential Set 3 (*Figure 2—figure supplement 4*).

## Stability guidance improves causal variant recovery in SuSiE

To explore the applicability of stability guidance beyond PICS, we developed a similar procedure that runs the fine-mapping algorithm SuSiE (*Wang et al., 2020*) on multiple slices and returns a stable variant. We call this approach *Stable SuSiE*. We then compared Stable SuSiE against SuSiE applied to residualized phenotypes (*Top SuSiE*), analogous to our comparisons for PICS earlier. First, similar to PICS, we generally observe no significant differences in performance between Stable and Top SuSiE, although empirically more causal variants are recovered by Top SuSiE in larger SNR simulation settings (*Figure 2C*), while more causal variants are recovered by Stable SuSiE in lower credible sets (*Figure 2—figure supplement 3*; see also *Figure 2—figure supplements 16–18*). Similar to PICS, we also observed matching fractions increasing with SNR in Potential Set 1 (*Appendix 12—table 3*). Next, comparing matching vs non-matching variants between the approaches, we observe a boost in causal variant recovery similar to that observed in PICS (*Figure 2D*), which persisted even when we stratified results by simulation parameters (*Figure 2—figure supplements 19 and 21*, *Figure 2—figure supplement 23*). However, one difference is that, unlike in PICS, this boost was unique to Credible Set 1—for the other two credible sets, improvements in causal variant recovery were observed only when SNR is high, with non-matching Stable SuSiE variants recovering at least one causal variant with the highest frequency in low SNR scenarios (*Figure 2—figure supplement 4*).

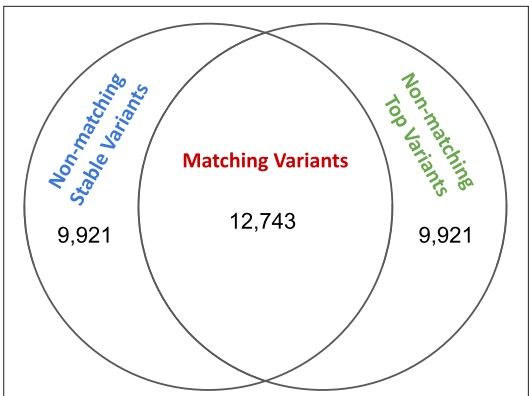

**Figure 3.** Venn diagram showing the number of matching and non-matching variants for Potential Set 1 in GEUVADIS fine-mapped variants.

The online version of this article includes the following figure supplement(s) for figure 3:

**Figure supplement 1.** Matching GEUVADIS Top vs Stable SNP posterior probabilities.

**Figure supplement 2.** Non-matching GEUVADIS Top vs Stable SNP posterior probabilities.

## Other analyses

Our key findings from the simulation study are as follows. Stability guidance, on its own, does not significantly outperform nor underperform other standard approaches at causal variant recovery. However, prioritizing variants that agree between stability-guided fine-mapping and standard fine-mapping can significantly boost causal variant recovery. In the Supplement, we also describe findings from investigations into the impact of including more potential sets on matching frequency and causal variant recovery (Appendix 2, with *Figure 2—figure supplements 5–7* discussed), the differences in PP between non-matching variants (Appendix 3, with *Figure 2—figure supplements 8 and 9* discussed), the relative performance of SuSiE and Stable PICS (Appendix 4, with *Figure 2—figure supplements 10–12* discussed), and interpreting matching variants with very low (stable) PP (Appendix 11).

## Stable variants frequently do not match top variants in GEUVADIS

Having identified scenarios that favor stability guidance, as well as strategies for using stability guidance to recover causal variants, we now turn to analysis of real gene expression data from GEUVADIS. As a preliminary analysis, when interrogating if the residualization and stability-guided approaches produce the same candidate causal variants, we find that for Potential Set 1, which corresponds to the set typically reported in fine-mapping studies, 56.2% of genes had matching top and stable variants (see *Figure 3*). Moving down potential sets, we find less agreement (Potential Set 2: 36.2%, Potential Set 3: 25.6%), providing evidence that as the marginal association of a variant with the expression phenotype decreases, the two approaches prioritize different signals when searching for putatively causal variants. This result is most consistent with our simulations using $S = 3$ causal variants and low SNR ($\phi = 0.05$), where we observed 56.5%, 37%, and 24.5% matching top and stable variants in Potential Sets 1, 2, and 3, respectively.

Because our simulations demonstrate that prioritizing matching variants boosts causal variant recovery, we proceeded with comparing functional impact of matching stable and top variants against non-matching variants. For the rest of this section, we report only results for Potential Set 1, given that it corresponds to the set typically reported in fine-mapping. Results for the other potential sets are provided in Appendices 7 and 8.

### Matching vs non-matching variants

For each gene, our algorithm finds the top eQTL variant and the stable eQTL variant, which may not coincide. We thus run (one-sided) unpaired Wilcoxon tests on matching and non-matching sets of variants to detect significant functional enrichment of one set of variants over the other. We find that the top variants that are also stable for the corresponding gene ($N_{\text{match}} = 12743$ maximum annotatable) score significantly higher in functional annotations than the top variants that are not stable ($N_{\text{non-match}} = 9921$ maximum annotatable). Notably, 361 out of 378 functional annotations report one-sided greater p-values <0.05 for the matching (i.e., both top and stable) variants after correcting for multiple testing using the Benjamini–Hochberg (BH) procedure, with many of these annotations measuring magnitudes of functional impact or functional enrichment (e.g., Enformer perturbation scores, FATHMM.XF score). Among the 17 remaining functional annotations, none has significantly lower scores for the matching variants. (Appendix 7 lists these functional annotations in detail.) We

also find that empirically, the matching variants tend to have greater agreement in posterior probabilities than non-matching variants (*Figure 2—figure supplements 1 and 2*).

As an example of a significant functional annotation, consider raw CADD scores (*Rentzsch et al., 2019*)—a higher value of which indicates a greater likelihood of deleterious effects. Out of the 22,559 genes for which both the top and the stable variant are annotatable, looking at the distribution of top variant scores, the one corresponding to the 12,685 genes with matching top and stable variants stochastically dominates the one corresponding to the remaining 9874 genes (*Figure 4A*). This relationship is more pronounced when we inspect PHRED-scaled CADD scores, where we apply a sliding cutoff threshold for calling variant deleteriousness (i.e., potential pathogenicity—see *Rentzsch et al., 2019*). We find that a greater proportion of matching variants than of either non-matching variant is classified as deleterious under the typical range of deleteriousness cutoffs (*Figure 4B*).

Another example demonstrating significant functional enrichment of matching variants over non-matching variants is the perturbation scores on H3K9me3 ChIP-seq peaks, as predicted by the Enformer (*Avsec et al., 2021*). Out of the 6364 genes for which the distance of both the top and the stable variant to the TSS are within the Enformer input sequence length constraint, looking at the distribution of top variant scores, the one corresponding to the 4491 genes with matching top and stable variants stochastically dominates the one corresponding to the remaining 1873 genes (*Figure 4C,D*). This relationship is true regardless of whether perturbation scores are calculated from an average of input sequences centered on the gene TSS and its two flanking positions (*Figure 4C*), or from input sequences centered on the gene TSS only (*Figure 4D*). (See Appendix 6 for details on Enformer annotation calculation.)

Similarly, amongst all stable variants, those variants that match the top variant are significantly more enriched in functional annotations: 363 functional annotations report one-sided greater BH-adjusted p-values <0.05 for the matching variants, and none of the remaining 15 annotations present significant depletion for the matching variants set.

## Top vs stable variants when they do not match

Focusing on the genes for which the top and stable variants are different, we run (one-sided) paired Wilcoxon tests to detect significant functional enrichment of one set of variants over the other. We find in general that for some genes, stable variants can carry more functional impact than top variants, and for other genes, top variants carry more functional impact—although neither of these patterns is statistically significant genome-wide after multiple testing correction using the BH procedure. For example, for raw CADD scores, out of the 9874 genes for which the top and the stable variants do not match, 4906 genes have higher scoring stable variants than top variants, whereas 4968 genes have higher scoring top variants than stable variants (*Figure 5A*). Looking at the accompanying PHRED-scaled CADD scores (CADD PHRED) for Potential Set 1, when applying a sliding cutoff threshold for calling variant deleteriousness, we find that even though a higher fraction of genes have top but not stable variant classified as deleterious, the difference is usually not significant (*Figure 5B*). *Figure 5B* also demonstrates that no matter how the cutoff is chosen, there are genes for which the top variant is not classified as deleterious while the stable variant is. Taken together, these observations suggest that the stability-guided approach can sometimes be more useful at identifying variants of functional significance, and in a broad sense, both fine-mapped variants should be equally prioritized for potential of carrying functional impact.

Because our comparison between the top and stable variants yielded no significant functional enrichment of one over the other, we investigate whether external factors—for example, the posterior probability of the variants—might moderate the relative enrichment of one variant over the other (i.e., we perform *trend analysis*—see Section for a list of all moderators). Here, we find that all but one of the moderators considered—namely, the Posterior Probability of Top Variant (see *Table 2*)—did not produce any significant trends for Potential Set 1. There is a small but significant positive correlation (Pearson's $r = 0.05$) between the posterior probability of the top variant and the difference between the stable variant and top variant FIRE scores (*Ioannidis et al., 2017*), and a small but significant negative correlation (Pearson's $r = -0.07$) between the posterior probability of the top variant and the difference between the stable variant and top variant Absolute Distance to Canonical TSS. (Details are reported in Appendix 8.)



**Figure 4.** Distribution of computational VEP scores across matching and non-matching variants. *Top row*. CADD scores. (**A**) Empirical cumulative distribution functions of raw CADD scores of matching and non-matching variants across all genes, for Potential Set 1. Non-matching variants are further divided into stable and top variants, with a score lower threshold of 1.0 and upper threshold of 5.0 used to improve visualization. (**B**) For a deleteriousness cutoff, the percent of (1) all matching variants, (2) all non-matching top variants, and (3) all non-matching stable variants, which are classified as deleterious. We use a sliding cutoff threshold ranging from 10 to 20 as recommended by CADD authors. For each value along the x-axis, 95% confidence intervals for point estimates on the y-axis were obtained using the Sison-Glaz method for constructing multinomial distribution standard errors (R command DescTools::MultinomCI(...)). *Bottom row*. Empirical cumulative distribution functions of perturbation scores of Enformer-predicted H3K27me3 ChIP-seq track. Score upper threshold of 0.015 and empirical CDF lower threshold of 0.5 used to improve visualization. (**C**) Perturbation scores computed from predictions based on centering input sequences on the gene TSS as well as its two flanking positions. (**D**) Perturbation scores computed from predictions based on centering input sequences on the gene TSS only.

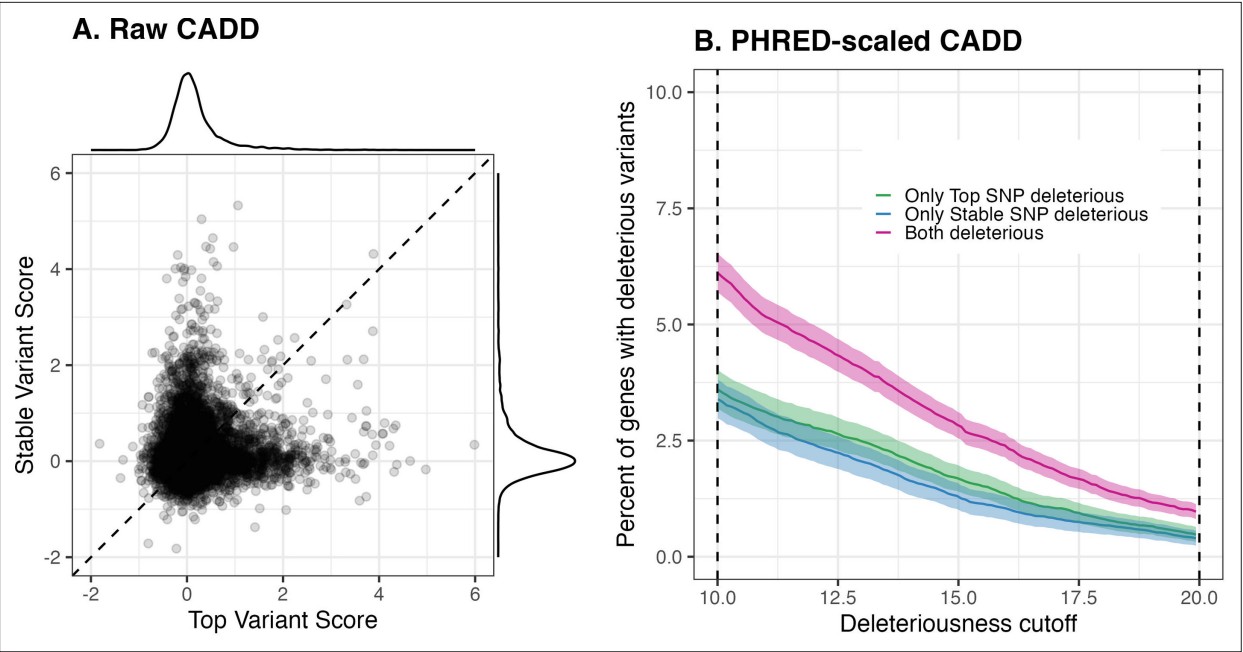

**Figure 5.** Comparison of CADD scores across non-matching top and stable variants. (**A**) Paired scatterplot of raw CADD scores of both top and stable variant for each gene, for Potential Set 1. (**B**) Percent of genes that are classified as (1) having deleterious top variant only, (2) having deleterious stable variant only, and (3) having both top and stable variant deleterious, using a sliding cutoff threshold ranging from 10 to 20 as recommended by CADD authors.

## Additional comparisons

We perform various *conditional analyses* to evaluate whether additional restrictions to characteristics of fine-mapping outputs may boost the power of either approach over the other. Such characteristics include (1) the positive posterior probability support (i.e., how many variants reported a positive posterior probability from fine-mapping), and (2) the posterior probability of the top or stable variant. Results from our conditional analysis of (1) are reported in Appendix 9. As an example, for (2) we further perform a comparison between top and stable variants, by restricting to genes where the posterior probability of the top variant or the stable variant exceeds 0.9. Such restriction of valid fine-mapped variant-gene pairs is useful in training variant effect prediction models requiring reliable positive and negative examples, as seen in *Wang et al., 2021*. We find that for genes where the top variant posterior probability exceeds 0.9, there is no significant enrichment of the top variant over the stable variant across the 378 annotations considered (all BH-adjusted p-values exceed 0.05). Interestingly, when focusing on genes where the stable variant posterior probability exceeds 0.9, FIRE scores

**Table 2.** List of six moderating factors considered.

| Moderator | Quantity/statistic computed |
|---|---|
| (1) Degree of Stability | No. subpopulations for which stable variant has positive probability |
| (2) Population Diversity | Maximum of pairwise allele frequency difference between subpopulations for which stable variant has positive posterior probability |
| (3) Population Differentiation | Maximum $F_{ST}$ between subpopulations for which stable variant has positive posterior probability |
| (4) Inclusion of Distal Subpopulations (Top) | Whether or not the top variant also had positive probability in Yoruban subpopulation when the stability-guided approach was used |
| (5) Inclusion of Distal Subpopulations (Stable) | Whether or not the stable variant had positive probability in Yoruban subpopulation when the stability-guided approach was used |
| (6) Degree of Certainty of Causality Using Residualization Approach | Posterior probability of top variant |

of the top variant are significantly larger than the stable variant (BH-adjusted p-value $= 1 \times 10^{-6}$). Detailed results are reported in Appendix 10.

## Discussion

We have shown that a stability-guided approach complements existing approaches to detect biologically meaningful variants in genetic fine-mapping. Through various statistical comparisons, we have found that prioritizing the agreement between existing approaches and a stability-guided approach enhances the functional impact of the fine-mapped variant. Incorporating stability into fine-mapping also provides an adjuvant approach that helps discover variants of potential functional impact in case standard approaches fail to pick up variants of functional significance. Our findings are consistent with earlier reports of stable discoveries having the tendency to capture actual physical or mechanistic relationships, potentially making such discoveries generalizable or portable (*Basu et al., 2018*).

The link between stability and generalizability is not new. In the machine learning and statistics literature, it has been shown that stable algorithms provably lead to generalization errors that are well-controlled (*Bousquet and Elisseeff, 2002*, Theorem 17, p. 510). Furthermore, in certain classes of algorithms (e.g., sparse regression), it has been demonstrated empirically that this mathematical relationship is explained precisely by the stable algorithm removing spurious discoveries (*Lim and Yu, 2016*).

The stability-guided approach is also distinct from other subsampling approaches, such as the bootstrap (*Efron and Tibshirani, 1994*). First, the bootstrap is more often deployed as a method for calibrating uncertainty surrounding a prediction, which is not the objective of our method. Next, in settings where the bootstrap is deployed as a type of perturbation against which a prediction or estimand is expected to be stable (*Basu et al., 2018*), a stability threshold is implicitly needed and would require tuning to be chosen (e.g., in *Basu et al., 2018* this threshold was chosen to be 50%). Here, we leverage interpretable existing external annotations to define the perturbation against which we expect the fine-mapped variant to be stable—in other words, the user relates what is meant by the fine-mapped variant being stable to a biologically meaningful concept like 'portable across environmentally and LD pattern-wise heterogeneous populations'.

The last sentence in the preceding paragraph suggests there are teleological similarities between the stability-guided approach and meta-analysis approaches (*Turley et al., 2021*). We emphasize that meta-analyses rely on already analyzed cohorts, thereby implicitly assuming that *within-cohort* heterogeneities have been sufficiently accounted for prior to the reporting of findings for that cohort. The stability-guided approach, however, is relevant to the *cohort-specific analysis itself*, where existing approaches may present methodological insufficiencies resulting in inflated false discoveries. In other words, whereas the goal of meta-analysis may be stated as identifying consistent hits across cohorts while also assuming that findings specific to each cohort are reliable, the goal of a stability-guided approach is to search for consistent signals despite the presence of potential confounders within a single cohort. Our analysis has focused on comparing the stability-guided approach against residualization, a convenient and popular approach to account for population structure, which reflects this difference in purpose.

Related to the previous point, while revising our work, we came across a few recent papers that developed multi-ancestry fine-mapping algorithms to investigate causal variant heterogeneity across ancestries. *Gao and Zhou, 2024* and *Yuan et al., 2024* found that complex traits have both shared and non-negligible ancestry-specific causal signals, although the proportion of trait-specific causal variants that are ancestry-specific is probably low (*Hu et al., 2025*; *Shi et al., 2021*). On the other hand, for molecular quantitative traits, which include RNA sequencing reads that we have studied in this work, *Lu et al., 2025* reported highly correlated effect sizes in *cis* QTLs across ancestries, with effect heterogeneity concentrated at predicted loss-of-function-intolerant genes. These findings are all consistent with our goal of leveraging stability guidance to identify generalizable causal signals and also imply that stability guidance would be particularly useful for molecular quantitative trait fine-mapping.

There are several limitations to our present work. First, we have chosen to focus on a non-parametric approach to fine-mapping, but multiple parametric fine-mapping approaches have been extended to incorporate cross-population heterogeneity (e.g., *LaPierre et al., 2021*; *Lu et al., 2022*; *Wen et al., 2015*). While our present work is a proof-of-concept of the applicability of the stability

to genetic fine-mapping, we believe that future work focusing on comparing functional impact of variants prioritized by our stability-driven approach or these parametric methods will shed more light on the efficacy of the stability principle at detecting generalizable biological signals. Our results for SuSiE on simulated gene expression data are promising in this regard. Second, our analyses assume that there is at most one *cis* eQTL prioritized by each potential set of PICS. This is owing to the implicit assumption that all other variants in high LD with the causal variant are simply tagging it, though we acknowledge the possibility of relaxing this assumption. Understanding the impact of this relaxation requires extensive investigation; hence, we defer it to future work. Third, while we have relied on numerous computational and experimental annotations to evaluate the functional impact of our fine-mapped variants, some computational variant effect predictors may themselves be biased owing to the lack of ancestral diversity of training data. Recent work has found that deep learning-based gene expression predictors—despite outperforming traditional statistical models at inferring regulatory tracks—explain surprisingly little of gene expression variation across ancestries (*Huang et al., 2023*) and yield limited power at predicting individual gene expression levels more broadly (*Sasse et al., 2023*). While this complicates the use of such predictors in their present form as strong evidence of functional impact, it is more constructive to understand their shortcomings and develop strategies to fine-tune their predictions for future use as trustworthy functional impact metrics—an emerging desideratum in the era of computational biomedicine (*Barbadilla-Martínez et al., 2025*; *Benegas et al., 2025*; *Katsonis et al., 2022*; *Livesey et al., 2025*).

In closing, while our work explores stability to subpopulation perturbations where subpopulations are defined by ancestry, we emphasize that our stability-guided slicing methodology is applicable to all settings where meaningful external labels are available to the data analyst. For instance, environmental or geographical variables, which are well-recognized determinants of some health outcomes (*Abdellaoui et al., 2022*; *Favé et al., 2018*) and arguably better measure potential confounders than ancestry labels, can be the basis on which slices are defined in the stability-guided approach. As the barriers to access larger biobank-scale datasets, which contain such aforementioned variables, continue to be lowered, we expect stability-driven analyses conducted on such data and relying on carefully defined slices will help users better understand genetic drivers of complex traits. Given its utility in our present work, we believe that the stability approach in precision medicine may find uses beyond genetic fine-mapping and few other biological tasks previously studied, ultimately empowering the discovery of veridical effects not previously known.

## Materials and methods

We use publicly available GEUVADIS B-lymphocyte RNA-seq measurements from 445 individuals (*Lappalainen et al., 2013*), whose genotypes are also available from the 1000 Genomes Project (*Auton et al., 2015*). The 445 individuals come from five populations: Tuscany Italian (TSI, $N = 91$), Great British (GBR, $N = 86$), Finnish (FIN, $N = 92$), Utah White American (CEU, $N = 89$), and Yoruban (YRI, $N = 87$). The mRNA measurements are normalized for library depth, expression frequency across individuals as well as PEER factors, as reported in the GEUVADIS Project (here). Of the available normalized gene expression phenotypes, only those with non-empty sets of variants lying within 1 Mb upstream or downstream of the canonical transcription start site were kept for (*cis*) fine-mapping. This process yielded 22,664 genes used in all subsequent analyses involving the PICS fine-mapping algorithm.

### Probabilistic identification of causal SNPs

We implement PICS (*Farh et al., 2015*), which is based on an earlier genome-wide analysis identifying genetic and epigenetic maps of causal autoimmune disease variants (*Farh et al., 2015*).

We assume that $\mathbf{X}$ and $\mathbf{y}$ are the $N \times P$ locus haplotype (or genotype) matrix and the $N \times 1$ vector of phenotype values, respectively, with $\mathbf{y}$ possibly already adjusted by relevant covariates. For consistency of exposition, let the SNPs be named $A_1, \ldots, A_P$. Recall the goal of fine-mapping is to return information about which SNP(s) in the set $\{A_1, \ldots, A_P\}$ is (are) causal, given the input pair $\{\mathbf{X}, \mathbf{y}\}$ and possibly other external information such as functional annotations or, more generally, prior biological knowledge.

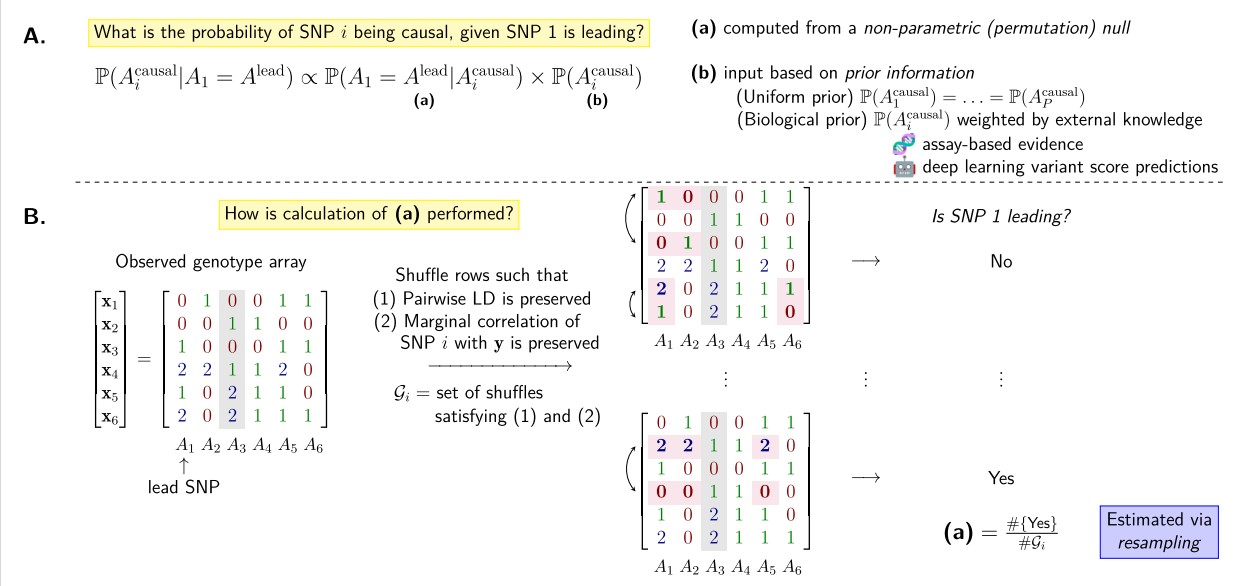

**Figure 6.** Visual summary of the PICS algorithm described in Probabilistic Identification of Causal SNPs. (**A**) Breakdown of the calculation of the probability of a focal SNP $A_i$ being causal. (**B**) Illustration of the permutation procedure used to generate the null distribution. An example $N \times P$ genotype array with $N = P = 6$ is used, with two valid row shuffles, or permutations, of the original array shown. Entries affected by the shuffle are highlighted, as is the focal SNP ($A_3$).

## Overview

PICS is a Bayesian, non-parametric approach to fine-mapping. Given the observed patterns of association at a locus, and furthermore not assuming a parametric model relating the causal variants to the trait itself, one can estimate the probability that any SNP is causal by performing permutations that preserve its marginal association with the trait as well as the LD patterns at the locus.

To see how this is accomplished, suppose that $A_1$ is the lead SNP. We are interested in $\mathbb{P}(A_i^{\text{causal}}|A_1 = A^{\text{lead}})$, the probability that $A_i$ is causal, for each $i \in [P]$. By Bayes' theorem,

$$\mathbb{P}(A_i^{\text{causal}}|A_1 = A^{\text{lead}}) \propto \mathbb{P}(A_1 = A^{\text{lead}}|A_i^{\text{causal}}) \times \mathbb{P}(A_i^{\text{causal}}). \tag{1}$$

Focusing on the focal SNP $i$, permute the rows of $\mathbf{X}$ such that the association between the focal SNP and the trait $\mathbf{y}$ is invariant; see Appendix 1 for a concrete mathematical description of this set of constrained permutations. Then the first term on the RHS of **Equation 1**, $\mathbb{P}(A_1 = A^{\text{lead}}|A_i^{\text{causal}})$, is estimated by the proportion of all permutations where $A_1$ emerges as the lead SNP. The second term, $\mathbb{P}(A_i^{\text{causal}})$, is the prior probability of the focal SNP being causal, which the user can choose based on prior knowledge. The default setting in PICS is $\mathbb{P}(A_1^{\text{causal}}) = \ldots = \mathbb{P}(A_P^{\text{causal}})$. **Figure 6** provides a visual summary of the method.

By running the permutation procedure across all SNPs in the locus, one obtains $\mathbb{P}(A_i^{\text{causal}}|A_1 = A^{\text{lead}})$ for each $i$. These posterior probabilities are then normalized so that $\sum_{i=1}^{N} \mathbb{P}(A_i^{\text{causal}}|A_1 = A^{\text{lead}}) = 1$.

Finally, the above procedure is performed after restricting the set of all SNPs to only those with correlation magnitude $|r| > 0.5$ to the lead SNP.

## Algorithm

Based on the computation of posterior probabilities described above (**Equation 1**), the full PICS algorithm returns putatively causal variants as follows.

The last line of Algorithm 1 returns a list of posterior probability vectors corresponding to each potential set. For our work, we set $|r| = 0.5$, $C = 3$, $R = 500$ and $\mathbf{p}_0 = (1/P, \ldots, 1/P)$ throughout implementations of Algorithm 1.

---

Algorithm 1. PICS.

---

1: **Input:** Individual-by-genotype array $\mathbf{X}_{N \times P}$, phenotype array $\mathbf{y}_{N \times 1}$, LD threshold $|r|$, resampling number $R$, number of potential sets $C$, prior causal probabilities $\mathbf{p}_0{}_{P \times 1}$.

2: $c \leftarrow 1$

3: **while** $c \leqslant C$ **do**

4: $M \leftarrow$ No. columns of $\mathbf{X}$

5: Identify lead SNP, $\ell \in \{1, \ldots, M\}$

6: Identify neighboring SNPs, $\mathcal{N}_\ell^c = \{k \in [M] : \mathrm{cor}(A_k, A_\ell)^2 > r^2\}$.

7: **for** $j \in \mathcal{N}_\ell^c$ **do**

8: Compute $\widehat{p}_j = \mathbb{P}(A_j^{\mathrm{causal}} | A_\ell = A^{\mathrm{lead}})$ as described below **Equation 1**. Use $R$ permutations.

9: **end for**

10: Record vector of posterior probabilities, $\overrightarrow{p^c} = [\widehat{p}_j : j \in \mathcal{N}_\ell^c]$

11: $\mathbf{X} \leftarrow \mathbf{X}[:, \{1, \ldots, M\} \setminus \mathcal{N}_\ell^c]$

12: $c \leftarrow c + 1$

13: **end while**

14: **Output:** $[\overrightarrow{p^c} : c \leqslant C]$ (a list, where component $c$ corresponds to Potential Set $c$)

---

As mentioned in the Introduction, we apply two different approaches to implementing Algorithm 1. One is guided by stability and will be introduced shortly in Incorporating stability. The other, which we now describe, is based on regressing out confounders, i.e., residualization. In implementing the residualization approach, we regress the top five principal components, obtained from the genotype matrix $\mathbf{X}$, from the gene expression phenotype, $\mathbf{y}$. We choose five principal components based on the elbow method, an approach described in **Brown et al., 2018**. This yields residuals $\mathbf{y}^r$ (**Figure 1A**), which we use as the input phenotype array in Algorithm 1. We subsequently report the variant with the highest posterior probability in each potential set as the putatively causal variant for that potential set. This variant is referred to as the top variant.

We remark that there are other ways of using potential confounders in variable selection, such as inclusion as covariates in a linear model. Because our algorithm explicitly avoids assuming a linear model, we have chosen the residualization approach just described.

## Incorporating stability

Our stability-guided approach to implementing PICS follows the steps outlined below. Let $(\mathbf{X}, \mathbf{y})$ be the genotype array and gene expression array. Assume that there are $K$ subpopulations making up the dataset, and let $E_k$ denote the set of row indices of $\mathbf{X}$ corresponding to individuals from subpopulation $k$. (In our present work, $K = 5$. The five subpopulations are Utahns ($N_{\mathrm{CEU}} = 89$), Finns ($N_{\mathrm{FIN}} = 92$), British ($N_{\mathrm{GBR}} = 86$), Toscani ($N_{\mathrm{TSI}} = 91$), and Yoruban ($N_{\mathrm{YRI}} = 87$).)

1. Run Algorithm 1 on the pair $(\mathbf{X}, \mathbf{y})$. Obtain a list of posterior probability vectors, $\mathcal{L} = [\overrightarrow{p^c} : c \leqslant C]$. Recall that $C$ is the number of potential sets.

2. For each subpopulation $k = 1, \ldots, K$, run Algorithm 1 on the pair $(\mathbf{X}|_{E_k}, \mathbf{y}|_{E_k})$. Obtain $\mathcal{L}^{(k)} = [\overrightarrow{p_k^c} : c \leqslant C]$ for each $k$.

3. (Stability-guided choice of putatively causal variants) Collect the probability vectors in lists $\mathcal{L}$ and $\mathcal{L}^{(k)}$ ($k \in [K]$). Operationalizing the principle that a stable variant has positive probability across multiple slices, we pick causal variants as follows. For potential set $c$, pick the variants that have (1) positive probability in $\overrightarrow{p^c}$ in Step 1, and moreover (2) have positive probability in the most number of probability vectors; call this set of variants $\mathcal{S}_c$ (note $\mathcal{S}_c$ is a subset of the support of $\overrightarrow{p^c}$ for Potential Set $c$). Among members of $\mathcal{S}_c$, select the variant that had the highest posterior probability in $\overrightarrow{p^c}$.

In other words, the stability-guided approach reports the variant that not only appears with positive probability in the most number of subsets including the pooled sample, but also has the largest probability in the pooled sample. This variant is referred to as the stable variant.

---

To illustrate the last step, suppose there are $K = 2$ subpopulations, $C = 1$ potential set, and $P = 5$ SNPs. Let the outputs from Steps 1 and 2 be

$$\mathcal{L}: \qquad \overrightarrow{p^1} = (0.45, 0.43, 0.02, 0.10, 0.00)$$
$$\mathcal{L}^{(1)}: \qquad \overrightarrow{p_1^1} = (0.00, 0.40, 0.10, 0.25, 0.25)$$
$$\mathcal{L}^{(2)}: \qquad \overrightarrow{p_2^1} = (0.20, 0.60, 0.00, 0.10, 0.10),$$

In this example, the second and fourth variants have positive probability in the most number of probability vectors ($\mathcal{S}_1 = \{A_2, A_4\}$). Among $A_2$ and $A_4$, $A_2$ has a higher posterior probability (i.e., 0.43) in $\overrightarrow{p^1}$, so the stable variant reported is $A_2$. As a side remark, the posterior probability vector $\mathcal{L}$ does not generally agree with the posterior probability vector computed using the residualization approach, because the latter is computed by running Algorithm 1 on the pair $(\mathbf{X}, \mathbf{y}^r)$ rather than $(\mathbf{X}, \mathbf{y})$, as described earlier.

## Stability-guided SuSiE

We apply the stability principle to SuSiE (*Wang et al., 2020*) in a similar manner as described above. Concretely, we run SuSiE on the pair $(\mathbf{X}, \mathbf{y})$, before running it again on each subpopulation $k = 1, \ldots, K$. Collecting the posterior inclusion probability (PIP) vectors across each subpopulation and the entire sample, we then keep only variants that have posterior probability at least 1/(no. variants included), owing to SuSiE tending to return nonzero probabilities to all variants included in the fine-mapping. These 'good' variants (i.e., variants with PIP at least as large as the prior inclusion probability) are analogous to the 'variants with positive probability' in PICS. The remaining steps are identical to stability-guided PICS.

## **Simulation study and evaluation details**

### Main simulation study

We simulate 2400 synthetic gene expression data, which vary by two parameters: $C$, the number of causal variants, and $\phi$, the proportion of variance in $\mathbf{y}$ explained by the genotypes $\mathbf{X}$. Phenotypes are simulated as follows.

1. For a gene, whenever its canonical transcription start site (TSS) is available, restrict the genotype matrix $\mathbf{X}$ to only variants lying within 1 Mb upstream and downstream. Else, use variants spanning the entire autosome associated with that gene. This ensures that *cis* eQTLs are used in simulation, for the most part.
2. Sample the indices of the $C$ causal variants, $\mathscr{C}$, uniformly at random from $\{1, \ldots, p\}$, where $p$ is the number of variants included in $\mathbf{X}$ from Step 1.
3. For each $j \in \mathscr{C}$, independently draw $b_j \sim N(0, 0.6^2)$ and, for all $j \notin \mathscr{C}$, set $b_j = 0$.
4. Set $\sigma^2$ to achieve the desired proportion of variance explained $\phi$.
5. Generate the phenotype by drawing $\mathbf{y} \sim N(\mathbf{X}\mathbf{b}, \sigma^2 \mathbf{I}_{n \times n})$.

After gene expression data were simulated, we ran three versions of PICS: the stability-guided version, the residualization version, and an uncorrected version, where no correction for population stratification or stability guidance was applied. To investigate the broader applicability of the stability approach, we also ran two versions of SuSiE (*Wang et al., 2020*), a popular fine-mapping method relying on variational approximation techniques. The first version runs SuSiE on residualized phenotypes, while the second version incorporates stability in selecting the most likely causal variant(s). In both approaches, we rely on default parameters of SuSiE while setting the number of credible sets to return to 3 (i.e., L=3 in `susieR::susie`).

### Smaller simulation study

To investigate whether non-genetic confounding between ancestries in a multi-ancestry cohort can hinder the performance of fine-mapping algorithms that do not correct for potential confounding by ancestry, we simulate a smaller set of synthetic gene expression data. We select 10 genes from Chromosomes 1, 20, 21, and 22 and repeat the simulation steps as described above, up to Step 4. At Step 4, instead of choosing a single $\sigma^2$, we allow $\sigma^2$ to depend on the ancestry of the individual in

order to capture environmental heterogeneity. We specifically consider six scenarios that we classify as (environmental) variance heterogeneity and mean heterogeneity.

- *Variance heterogeneity.* Let $\boldsymbol{\alpha}(t) = (\alpha_{\text{TSI}}(t), \alpha_{\text{GBR}}(t), \alpha_{\text{FIN}}(t), \alpha_{\text{CEU}}(t), \alpha_{\text{YRI}}(t))$ be a vector of exponentiated proportions, where we define the exponentiated proportion of a subpopulation POP (POP $\in \{\text{TSI}, \text{FIN}, \text{GBR}, \text{CEU}, \text{YRI}\}$) as $\alpha_{\text{POP}}(t) = N_{\text{POP}}^t/(N_{\text{TSI}}^t + N_{\text{GBR}}^t + N_{\text{FIN}}^t + N_{\text{CEU}}^t + N_{\text{YRI}}^t)$. In Step 4, we define a vector of ancestry-specific exogenous variances, $\boldsymbol{\sigma^2}(t) = 5\sigma^2 \boldsymbol{\alpha}(t)$, where $\sigma^2$ is calculated as in Step 4 of the original simulation setup. We then generate phenotypes independently in Step 5 by conditioning on the individual's population membership (POP): $y_i \sim N(\boldsymbol{x}_i\mathbf{b}, \boldsymbol{\sigma^2}(t)(k))$, where $k$ is the index that corresponds to the population membership of individual $i$ and $\boldsymbol{x}_i$ denotes the genotypes of individual $i$. For example, if $t = 0$, $\boldsymbol{\alpha}(t) = \boldsymbol{\alpha}(0) = (1/5, 1/5, 1/5, 1/5, 1/5)$ and $\boldsymbol{\sigma^2}(t) = (\sigma^2, \sigma^2, \sigma^2, \sigma^2, \sigma^2)$, which reduces to our original simulation setup. The choice of $t > 0$ determines the heterogeneity of the exogenous variance among the subpopulations, with a larger $t$ producing higher heterogeneity. Here, we consider four choices of $t$: 8, 16, 128, and 256.
- *Mean heterogeneity.* Step 5 of the original simulation setup is modified to include an ancestry-dependent mean vector. We arbitrarily let GBR be the 'focal' population and generate phenotypes in two ways.
  - 'Smooth mean': After $\sigma^2$ is calculated in Step 4 of the original simulation setup, define the mean exogenous noise vector as $\boldsymbol{\mu} = (\mu_{\text{TSI}}, \mu_{\text{GBR}}, \mu_{\text{FIN}}, \mu_{\text{CEU}}, \mu_{\text{YRI}}) = (2\sigma, 0, \sigma, \sigma, 2\sigma)$. An individual $i$ has their phenotype simulated as $y_i \sim N(\boldsymbol{x}_i\mathbf{b} + \boldsymbol{\mu}(k), \sigma^2)$, where again $k$ is the index that corresponds to the population membership of individual $i$. This effectively increases the gene expression of all but GBR individuals in a positive direction with varying amounts of shift.
  - 'Spiked mean': After $\sigma^2$ is calculated in Step 4 of the original simulation setup, define $\boldsymbol{\mu} = (0, 2\sigma, 0, 0, 0)$ and simulate individual $i$'s phenotype as $y_i \sim N(\boldsymbol{x}_i\mathbf{b} + \boldsymbol{\mu}(k), \sigma^2)$. This effectively increases the gene expression of GBR individuals only in a positive direction, while the other subpopulation individuals share the same zero mean (as in the original simulation setup).

For brevity, we refer to these six scenarios by `t=8`, `t=16`, `t=128`, `t=256`, `|i-3|` and `i=3`.

## Evaluation

Several metrics have been used to evaluate Bayesian fine-mapping algorithms (see e.g., **Benner et al., 2016**; **Cui et al., 2024**; **Hormozdiari et al., 2015**; **Wang et al., 2020**; **Wen et al., 2016**; **Yang et al., 2023**), many of which compute credible sets and analyze their coverage. A challenge in using credible sets for our study is that the posterior probabilities returned by the PICS algorithm are not posterior inclusion probabilities in a strict sense: they do not measure the frequency with which a variant should be included in a *linear model* (**Griffin and Steel, 2021**; note PICS does not assume a model relating features to outcome). This complicates defining credible sets and coverage, so we follow **Mazumder, 2020** instead and evaluate performance of all algorithms by computing the signal recovery probability as a function of the signal to background noise (SNR). By the definition of $\phi$ (see **Wang et al., 2020** for details), we may compute SNR directly from $\phi$ as $\text{SNR} = \phi/(1 - \phi)$.

## Statistical comparison methodology

We rely on 378 external functional annotations to interpret biological significance of our variants, summarized in **Table 1**. Appendix 5 provides interpretation for the functional annotations, while Appendix 6 describes in greater detail how we generate annotations from Enformer predictions.

Our comparison of the top and stable variants is twofold. First, we evaluate the relative significance of the stable variant against the top variant by running paired Wilcoxon two-sample tests across all pairs of top and stable variants across all GEUVADIS gene expression phenotypes. We compute one-directional p-values in either direction to check for significant depletion or enrichment of the stable variant with respect to a particular annotation. Raw p-values are adjusted for false discovery rate control by applying the standard BH procedure (R command `p.adjust(…,method='BH')`) to p-values across all 378 annotations and all potential sets.

To investigate whether various external factors might moderate the differences in functional annotations, we next perform trend tests. Basically, for some external factor $F$, we run a trend test to see if values of $F$ are associated with attenuation or augmentation of differences in functional annotation of the top and stable variants. The list of all external factors $F$ is provided in **Table 2**.

For (1) Degree of Stability, we perform a Jonckheere–Terpstra test (R command `clinfun::jonckheere.test(…,nperm=5000)`). For (2) Population Diversity, (3) Population Differentiation, and (6) Degree of Certainty of Causality Using Residualization Approach, we perform a correlation test (R command `cor.test(…)`). For (4) Inclusion of Distal Subpopulations (Top) and (5) Inclusion of Distal Subpopulations (Stable), we perform an unpaired Wilcoxon test (R command `wilcox.test(…)`). Finally, for each moderator, we again compute one-directional p-values in either direction, before applying the BH procedure to all p-values across annotations and potential sets for that moderator only.

## Acknowledgements

This research is supported in part by an NIH grant R35-GM134922 and grant number CZF2019-002449 from the Chan Zuckerberg Initiative Foundation. We thank Carlos Albors, Gonzalo Benegas, Ryan Chung, Ziyue Gao, Iain Mathieson, and Yutong Wang for helpful discussions; members of the Yu Group at Berkeley for feedback on the work; and Aniketh Reddy for help with data processing.

## Additional information

### Competing interests
Lionel Chentian Jin: Lionel Chentian Jin is affiliated with McKinsey & Company. The author has no other competing interests to declare. The other authors declare that no competing interests exist.

### Funding

| Funder | Grant reference number | Author |
| --- | --- | --- |
| National Institutes of Health | R35-GM134922 | Yun S Song |
| Chan Zuckerberg Initiative | CZF2019-002449 | Yun S Song |

The funders had no role in study design, data collection, and interpretation, or the decision to submit the work for publication.

### Author contributions
Alan J Aw, Conceptualization, Software, Formal analysis, Investigation, Visualization, Methodology, Writing – original draft; Lionel Chentian Jin, Software, Visualization; Nilah Ioannidis, Supervision, Writing – review and editing, Conceptualization, Investigation, Methodology; Yun S Song, Supervision, Funding acquisition, Writing – review and editing, Conceptualization, Investigation, Methodology

### Author ORCIDs
Alan J Aw ⓘ https://orcid.org/0000-0001-9455-7878
Yun S Song ⓘ https://orcid.org/0000-0002-0734-9868

Reviewer #1 (Public review): https://doi.org/10.7554/eLife.88039.3.sa1
Reviewer #2 (Public review): https://doi.org/10.7554/eLife.88039.3.sa2
Author response https://doi.org/10.7554/eLife.88039.3.sa3

## Additional files

### Supplementary files
MDAR checklist

### Data availability
We provide the following data and scripts on the GitHub repository https://github.com/songlab-cal/StableFM (copy archived at *Aw, 2026*): fine-mapped variants with their functional annotations and

moderator variable quantities, code for reproducing Main Text figures in this manuscript and building our visualization app.

The following previously published dataset was used:

| Author(s) | Year | Dataset title | Dataset URL | Database and Identifier |
|---|---|---|---|---|
| Dermitzakis E, Kurbatova N, Lappalainen T | 2013 | RNA-sequencing of 465 lymphoblastoid cell lines from the 1000 Genomes Project (GEUVADIS) | https://www.ebi.ac.uk/biostudies/arrayexpress/studies/E-GEUV-1 | ArrayExpress, E-GEUV-1 |

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

## Appendix 1

### Permuting while preserving marginal correlation and LD

For a focal SNP $i$, its $N$ entries $(x_{1i}, \ldots, x_{Ni})$ take on only $D$ values, where $D$ is the ploidy of the data (for human genotypes $D = 2$ and for human haplotypes $D = 1$). Suppose $D = 2$ for exposition. For $d = 0, 1, 2$, let $I_d \subset [N]$ denote the indices of those elements of the allelic dosage vector $(x_{1i}, \ldots, x_{Ni})$ taking on value $d$. Then, any permutation $\sigma \in \mathfrak{S}_N$ of the rows of $\mathbf{X}$ that preserves marginal correlation with the phenotype and LD maps indices from $I_d$ to $I_d$, because

- Each distinct entry in the allelic dosage vector of the focal SNP is still assigned the same phenotype entry;
- Row shuffles do not change column-column covariances, the latter of which determines LD.

To be precise, $\sigma \in \mathfrak{S}_{|I_0|} \times \mathfrak{S}_{|I_1|} \times \mathfrak{S}_{|I_2|}$, a direct product of the three permutation groups corresponding to the different allelic dosages.

In practice, we approximate the permutation distribution by sampling permutations uniformly at random from the group defined above. We set the sampling number to 500 for all our fine-mapping experiments.

## Appendix 2

### Impact of including more potential sets on matching frequency and causal variant recovery

The number of credible or potential sets is a parameter in many fine-mapping algorithms. Focusing on stability-guided approaches, we consider how including more potential sets for stable fine-mapping algorithms affects both causal variant recovery and matching frequency in simulations. For the latter, we will specifically investigate whether including more potential sets in PICS and searching for matching variants across different potential sets for Top and Stable PICS will improve causal variant recovery.

#### Causal variant recovery

We investigate both Stable PICS and Stable SuSiE. Focusing first on simulations with one causal variant, we observe a modest gain in causal variant recovery for both Stable PICS and Stable SuSiE, most noticeably when the number of sets was increased from 1 to 2 under the lowest signal-to-noise ratio setting (SNR = 0.053, see *Figure 2—figure supplement 5*). For example, for Stable PICS, we observed 4 genes—ENSG00000203760.4 in Chr6, ENSG00000132912.8 in Chr5, and ENSG00000168614.12 and ENSG00000171502.10 in Chr1—where the most stable variant in Potential Set 2 was the causal variant, which we would not have recovered had we only told Stable PICS to return one potential set.

When we consider simulations with two and three causal variants, we observe modest gains when increasing from 1 to 2 sets (*Figure 2—figure supplement 6*) and from 2 to 3 sets (*Figure 2—figure supplement 7*). Bigger gains were observed for Stable SuSiE than for Stable PICS, especially from 1 to 2 sets, where Stable SuSiE saw larger increases in probability of recovering all causal variants across all SNR settings.

#### Matching frequency

In PICS, we assume that each potential set returns just one causal variant, because PICS explicitly groups variants with very high LD within the same potential set. Here, we ask whether non-matching variants between the same potential sets returned by Top and Stable PICS are owing to the lack of multiple causal variant modeling; in other words, whether non-matching variants would match across *different* Top PICs and Stable PICS potential sets.

Computing fractions of non-matching variants for which the stable variant matched the top variant of another potential set, we observed generally small matching fractions (see *Appendix 12—table 4*). The mean matching fraction between Potential Set 1 stable variant and a top variant in Potential Set 2 or 3 across all simulations was 11%, with larger matching fractions occurring between Potential Set 1 stable and Potential Set 2 top variants. Mean matching fractions for Potential Sets 2 and 3 stable variant and a different potential set top variant were 23% and 14%, with larger matching fractions occurring between Potential Set 2 stable and Potential Set 3 top variants and Potential Set 3 stable and Potential Set 2 top variants, respectively.

We next checked if these 'off-diagonal' matching variants are enriched in causal variants. Across all simulations, an off-diagonal matching Potential Set 1 stable variant recovered a causal variant 54% of the time, while for Potential Set 2 stable variant and Potential Set 3 stable variant the recovery percentages were 16% and 8%. These fractions are larger than if the stable variant did not match any top variant (24%, 5%, and 3% for Potential Sets 1, 2, and 3, respectively). However, these fractions are generally smaller when compared against causal variant recovery percentages for matching variants: we observed smaller fractions for Potential Set 1 (54% < 77%) and Potential Set 2 (16% < 20%), and only for Potential Set 3 did we observe a slightly larger fraction (8% > 6%).

Taken together, these findings demonstrate that it is helpful to condition on matching variants between 'off-diagonal' potential sets to recover causal variants in case the top and stable variants in the same potential set do not match, but such events occur infrequently and do not enrich for causal variants as much as do matching variants from the same potential sets. Therefore, in our real data analysis, we proceed with comparing matching variants from the same potential sets.

## Appendix 3

### Difference in posterior probability of non-matching variants

We investigate whether there are differences in posterior probabilities between non-matching variants in simulations. Focusing first on Potential Set 1 non-matching variants, we observe that posterior probabilities tend to be clustered around two regions: (1) similar posterior probabilities that are less than 0.5; and (2) low stable variant posterior probabilities (*Figure 2—figure supplement 9*). This clustering pattern differs from matching variants, which tend to have more similar and larger posterior probabilities (*Figure 2—figure supplement 8*). Moving down to Potential Sets 2 and 3, we observe a similar clustering around region (2) for non-matching variants, with the remaining points distributed more randomly within the rectangle [0,1] × [0,1] (*Figure 2—figure supplement 9*). Matching variants for these two potential sets remain more similar (lying close to $y = x$ diagonal of the rectangle, see *Figure 2—figure supplement 8*), suggesting that a salient feature of non-matching variants is a small stable variant posterior probability (despite possibly large top variant posterior probability).

## Appendix 4

### Comparing SuSiE to stable PICS

We investigate how PICS performs relative to SuSiE, a widely used fine-mapping algorithm, in simulations. Our comparison is between Stable PICS and Top SuSiE, the latter of which performs residualization on gene expression phenotypes to remove potential confounding by ancestry. Starting with simulations involving one causal variant, we observe slightly better causal variant recovery frequency for SuSiE at larger SNR, although the opposite is true at smaller SNR (*Figure 2—figure supplement 1*). Notably, PICS always recovers causal variants with probability at least 0.5. Moving onto simulations with two causal variants (*Figure 2—figure supplement 2*) and three causal variants (*Figure 2—figure supplement 3*), we observe similar patterns, where PICS recovers more causal variants at small SNR and SuSiE recovers more causal variants at large SNR. (Comparisons between Stable PICS and Stable SuSiE are summarized in *Figure 2—figure supplements 5–7*.)

## Appendix 5

### List of annotations

We measure the biological significance of a variant using a wide range of available functional annotations (see *Appendix 12—table 6*). The annotations cover potential biological activity in the local vicinity of the variant position along the genome, estimated model-based biological quantities (e.g., selection and conservation scores), and predicted effects of mutagenesis from the reference to the alternate allele at the variant.

## Appendix 6

### Generating annotations from enformer predictions

The Enformer (*Avsec et al., 2021*) is a sequence-based prediction model, which leverages the attention mechanism in a transformer neural network to capture long-range effects on gene regulation.

For Enformer predictions, we subset the 5313 ChIP-seq, DNase-seq, ATAC-seq, and CAGE tracks to only those relevant to the lymphoblastoid cell line (GM12878), the cell line with respect to which GEUVADIS gene expression phenotypes are measured. We also restrict to genes whose top and stable variants are within the 196,608 bp input length limit of the Enformer from the corresponding GEUVADIS gene's TSS. This restriction allows us to obtain Enformer predictions on three sequences: a *null sequence*, a sequence where the REF allele is replaced with the ALT allele at the top variant (*top sequence*), and a sequence where the REF allele is replaced with the ALT allele at the stable variant (*stable sequence*).

To compare the top and the stable variants with respect to a particular track, we take the predictions obtained from the three input sequences (null prediction, top prediction, stable prediction—a *triplet*), and compute the magnitude of change in predictions between both the top and null and the stable and null. We use the magnitude of change here, rather than the change itself, to capture the impact the mutation associated with the variant has on the track. Our Enformer functional annotations can thus be interpreted as measures of mutagenic impact on a track's prediction, without considering the directionality of the impact. We refer to these annotations as *perturbation scores*.

We compute perturbation scores in two ways. The first way is to center all input sequences on the transcription start site of the gene (occasionally referred to as 'TSS' in our work), thus allowing the model to measure the impact of mutagenesis at the (top or stable) variant on predicted gene expression profile for the sequence. The second way is to average the perturbation scores over three triplets, one centered on the gene TSS, and two centered on the flanking positions of the gene TSS (i.e., the two neighboring bins). The second way (occasionally referred to as 'AVE' in our work) accounts for errors arising from imprecise TSS positioning and the instability of the model to small changes in input—a technique used in *Karollus et al., 2023* and by the Enformer team (*Avsec et al., 2021*).

## Appendix 7

### Matching vs non-matching variant results

As described in the main text, we compare between one set of genes where the top and stable variants match and another set of genes where the top and stable variants disagree; comparisons are made between matching variants and one of the non-matching sets of variants (top or stable).

Concretely, for Potential Set 1, we compare between $N_{\text{match}} = 12743$ genes with matching variants and $N_{\text{non-match}} = 9921$ genes with non-matching variants; for Potential Set 2, we compare between $N_{\text{match}} = 8197$ genes with matching variants and $N_{\text{non-match}} = 14452$ genes with non-matching variants; and for Potential Set 3, we compare between $N_{\text{match}} = 5807$ genes with matching variants and $N_{\text{non-match}} = 16812$ genes with non-matching variants. Below, we summarize our findings.

#### Matching variants vs non-matching top variants

We observe the following significant trends: as mentioned in the main text, for Potential Set 1 361 functional annotations reported significantly higher scores for the matching variants, with all BH-adjusted p-values less than 0.05. The remaining $17 = 378 - 361$ annotations *not* exhibiting significantly higher scores are: Distance to Canonical TSS, CTCF Binding Enrichment, Enhancer Enrichment, Open Chromatin Enrichment, TF Binding Enrichment, Promoter Flanking Enrichment, CADD raw score, PHRED-normalized CADD score, SIFTVal, priPhyloP, mamPhyloP, verPhyloP, GerpS, LINSIGHT, Funseq2, ALoft, and FIRE.

For Potential Set 2, 132 functional annotations reported significantly higher scores for the matching variants. For Potential Set 3, 9 functional annotations reported significantly higher scores for the matching variants: B Statistic, percent GC content, and Enformer tracks `ENCFF776DPQ`, `ENCFF831ZHL`, `ENCFF601YET`, `ENCFF945XXY`, `ENCFF676UXN`, `ENCFF151LGF`, `ENCFF876DXW`. For all three potential sets, there were no functional annotations reporting significantly lower scores for the matching variants, even for annotations where a low score implies greater biological functionality (e.g., B Statistic).

#### Matching variants vs non-matching stable variants

We observe the following significant trends: as mentioned in the main text, for Potential Set 1, 363 functional annotations reported significantly higher scores for the matching variants, with all BH-adjusted p-values less than 0.05. The remaining 15 annotations *not* exhibiting significantly higher scores are: Distance to Canonical TSS, CTCF Binding Enrichment, Enhancer Enrichment, Open Chromatin Enrichment, TF Binding Enrichment, Promoter Flanking Enrichment, CADD raw score, PHRED-normalized CADD score, SIFTVal, mamPhyloP, verPhyloP, GerpS, LINSIGHT, ALoft, and FIRE.

For Potential Set 2, 359 functional annotations reported significantly higher scores for the matching variants (all BH-adjusted p-values <0.05), while only one functional annotation—Distance to Canonical TSS—reported significantly lower scores for the matching variants (BH-adjusted p-value $= 8 \times 10^{-4}$). For Potential Set 3, 259 functional annotations reported significantly higher scores for the matching variants (all BH-adjusted p-values <0.05), while only one functional annotation—Distance to Canonical TSS—reported significantly lower scores for the matching variants (BH-adjusted p-value $= 8 \times 10^{-4}$).

## Appendix 8

### Top variant vs stable variant results

As described in the main text, we compare, across all genes, the annotations of the top and the stable variant. We reported no significant differences in enrichment or trends in enrichment differences driven by moderators for Potential Set 1. For completeness, we report here significant results for all three potential sets.

### Paired analysis

After BH adjustment, there are no significant differences across all three potential sets using a threshold of $\alpha = 0.05$.

### Trend analysis

For the following moderators, we observe some significant trends.

- Inclusion of Distal Subpopulations (Top)
  For Potential Set 3, FIRE scores exhibited a greater difference between stable and top variant scores (i.e., stable score − top score) for genes with top variants not discovered in YRI than those with top variants discovered in YRI (one-sided BH-adjusted $p = 0.03$). So did TSS-centered perturbation scores for 121 Enformer tracks and AVE perturbation scores for 121 Enformer tracks. (These tracks and their corresponding BH-adjusted p-values are available here.)

- Posterior Probability of Top Variant
  For all potential sets, there are small but significant positive correlations between the posterior probability of the top variant and the difference between the stable variant FIRE score and the top variant FIRE score (Potential Set 1: Pearson's $r = 0.05$, BH-adjusted $p = 3 \times 10^{-5}$; Potential Set 2: Pearson's $r = 0.07$, BH-adjusted $p = 3 \times 10^{-14}$; Potential Set 3: Pearson's $r = 0.05$, BH-adjusted $p = 1 \times 10^{-7}$). For all potential sets, there are small but significant negative correlations between the posterior probability of the top variant and the difference between the stable variant Absolute Distance to TSS and the top variant Absolute Distance to TSS (Potential Set 1: Pearson's $r = -0.07$, BH-adjusted $p = 1 \times 10^{-10}$; Potential Set 2: Pearson's $r = -0.05$, BH-adjusted $p = 9 \times 10^{-7}$; Potential Set 3: Pearson's $r = -0.06$, BH-adjusted $p = 2 \times 10^{-12}$). For Potential Set 2, there are significant negative correlations between the posterior probability of the top variant and the difference between the stable variant and top variant Percent GC (Pearson's $r = -0.04$, BH-adjusted $p = 5 \times 10^{-5}$) and FATHMM-XF scores (Pearson's $r = -0.04$, BH-adjusted $p = 7 \times 10^{-4}$). For Potential Set 3, there is a significant negative correlation between the posterior probability of the top variant and the difference between the stable variant and top variant FATHMM-XF scores (Pearson's $r = -0.04$, BH-adjusted $p = 2 \times 10^{-5}$).

## Appendix 9

### Impact of positive posterior probability support on results

One key consideration in statistical fine-mapping is the number of variants possessing positive posterior probability, which we refer to as the *positive posterior probability support* (or *support*, for short). Indeed, a larger support may indicate greater uncertainty in the assignment of causal variant to the allele with largest posterior probability. To evaluate the potential utility of the stability-guided approach in settings where the residualization approach leads to large support, we repeat our analysis on two restricted sets of genes, namely (1) those genes where the top variant has support greater than 10; or (2) those genes where the top variant has support greater than 50.

#### Paired analysis

*Gene Set (1)*. For all Potential Sets, Distance to TSS is significantly smaller for top variant (all three BH-adjusted p-values $< 10^{-6}$). For Potential Set 3, FATHMM-XF score is significantly larger for stable variant (BH-adjusted $p < 10^{-6}$). For all Potential Sets, FIRE score is significantly larger for top variant (BH-adjusted p-values: Potential Set 1 = 0.02, Potential Set 2 = $2 \times 10^{-10}$, Potential Set 3 = $1.7 \times 10^{-4}$). *Gene Set (2)*. For all Potential Sets 2 and 3, FIRE score is significantly larger for top variant (both BH-adjusted p-values are 0.04).

#### Trend analysis

*Gene Set (1)*. For the following moderators, we observe some significant trends.

- Posterior Probability of Top Variant
  For Potential Sets 2 and 3, there is a significant negative correlation between the posterior probability of the top variant and the difference between the stable variant FIRE score and the top variant FIRE score (Potential Set 2: Pearson's $r = 0.08$, BH-adjusted $p = 1.9 \times 10^{-5}$; Potential Set 3: Pearson's $r = 0.06$, BH-adjusted $p = 0.003$). For Potential Set 2, there is a significant positive correlation between the posterior probability of the top variant and the difference between the stable variant's FATHMM-XF score and the top variant's FATHMM-XF score (Pearson's $r = -0.06$, BH-adjusted $p = 0.01$).

*Gene Set (2)*. For the following moderators, we observe some significant trends.

- Posterior Probability of Top Variant
  For Potential Set 2, TSS-centered perturbation scores for 8 Enformer tracks exhibited a positive correlation between the posterior probability of the top variant and the difference between stable and top variant perturbation scores. Track names (with empirical Pearson's $r$) are `ENCFF915DFR (-0.40)`, `ENCFF107LDM (-0.41)`, `ENCFF821PRO (-0.42)`, `ENCFF171MDW (-0.43)`, `ENCFF170NTY (-0.43)`, `ENCFF935KTD (-0.41)`, `ENCFF676GTP (-0.41)`, `ENCFF013ZOI (-0.42)`. Averaged perturbation scores for two Enformer tracks also exhibited negative correlation; these two tracks are `ENCFF107LDM (-0.40)` and `ENCFF170NTY (-0.40)`. All BH-adjusted p-values are 0.0495.

## Appendix 10

### Impact of posterior probability on results

We perform a comparison between top and stable variants, by restricting to genes where the posterior probability of the top variant or the stable variant exceeds 0.9. Specifically, we repeat our analysis on two restricted sets of genes, namely (1) those genes where the top variant reported a posterior probability exceeding 0.9; or (2) those genes where the stable variant reported a posterior probability exceeding 0.9. For reference, we plot in *Figure 3—figure supplement 2* the joint distribution of posterior probabilities of top and stable variants, across all genes for which the two fine-mapping approaches returned distinct variants.

#### Paired analysis

*Gene Set (1).* For Potential Set 3, Distance to TSS is significantly smaller for stable variant (BH-adjusted $p = 0.01$). *Gene Set (2).* For Potential Sets 1 and 2, FIRE scores of the top variant are significantly larger than the stable variant (BH-adjusted p-values: Potential Set 1 = $1 \times 10^{-6}$, Potential Set 2 = 0.02). For Potential Set 2, the distance to TSS of the top variant is significantly smaller (BH-adjusted $p = 3 \times 10^{-4}$). For Potential Set 3, FATHMM-XF scores of the stable variant are significantly larger (BH-adjusted $p = 0.006$).

#### Trend analysis

*Gene Set (1).* For the following moderators, we observe some significant trends.

- Inclusion of Distal Subpopulations (Top)
  For Potential Set 3, TSS-centered perturbation scores for nine Enformer tracks exhibited a higher difference between stable and top variant perturbation score for genes with top variants not discovered in YRI vs those discovered in YRI. Track names (with BH-adjusted unpaired Wilcoxon test p-values) are `ENCFF279CYY (0.03)`, `ENCFF417WYL (0.04)`, `ENCFF782WWH (0.03)`, `ENCFF629RRF (0.03)`, `ENCFF676GTP (0.03)`, `ENCFF984HLU (0.03)`, `ENCFF700YOH (0.03)`, `ENCFF848LJL (0.0477)`, `ENCFF038IYA (0.03)`. `Averaged perturbation scores for 10 Enformer tracks also a higher difference; these are ENCFF776DPQ (0.02)`, `ENCFF279CYY (0.02)`, `ENCFF629RRF (0.03)`, `ENCFF319YAI (0.04)`, `ENCFF676GTP (0.046)`, `ENCFF367WTF (0.04)`, `ENCFF984HLU (0.04)`, `ENCFF917YSR (0.03)`, `ENCFF613CYH (0.04)`, and `ENCFF700YOH (0.04)`.

*Gene Set (2).* For the following moderators, we observe some significant trends.

- Posterior Probability of Top Variant
  For Potential Set 3, there is a significant negative correlation between the posterior probability of the top variant and the difference between the stable variant Distance to TSS and the top variant Distance to TSS (Pearson's $r = -0.09$, BH-adjusted $p = 0.004$).

## Appendix 11

### On matching variants with very low (stable) posterior probability

In both simulations and analysis of GEUVADIS data, some matching variants have low stable posterior probability. To better interpret such variants, we look at allele frequency heterogeneity by ancestry, posterior probability support size, and (in Stable PICS) number of slices containing the variant. For simulated data, we additionally consider if such low posterior probabilities can be predictive of causal variant recovery. For GEUVADIS data, we compare functional enrichments of such variants relative to other matching variants.

#### Simulated gene expression

By defining a very low stable posterior probability as having a probability <0.01, we found two expression phenotypes that returned matching top and stable variants in Potential Set 1: `rs8008094` (using TSS metadata for gene `ENSG00000151413.12`) and `rs4758290` (using TSS metadata for gene `ENSG00000258659.1`). Both simulations involved two causal variants and >1000 background variants. Allele frequencies for both variants did not differ substantially across ancestry slices: $(f_{YRI}, f_{TSI}, f_{GBR}, f_{FIN}, f_{CEU}) = (0.408, 0.302, 0.198, 0.25, 0.208)$ for `rs8008094` with pooled allele frequency $f = 0.273$, while $(f_{YRI}, f_{TSI}, f_{GBR}, f_{FIN}, f_{CEU}) = (0.540, 0.527, 0.523, 0.571, 0.528)$ for `rs4758290` with pooled allele frequency $f = 0.538$. However, for `rs8008094`, the positive posterior probability support size was 47, whereas for `rs4758290`, the positive posterior probability support size was 7. Additionally, Top PICS returned a posterior probability of 0.359 for `rs8008094` and 0.846 for `rs4758290`. The latter variant also appeared in five slices for Stable PICS, while the former appeared in three slices. It turns out that `rs4758290` was one of the causal variants in the simulations whereas `rs8008094` was not.

Moving onto Potential Sets 2 and 3 matching variants, using the same cutoff, we identified five (`rs4687770`, `rs2834344`, `rs1125036`, `rs7963386`, `rs74080151`) and one (`rs1404862`) variant(s), respectively, that had low stable posterior probability. All but two simulations involved three causal variants—`rs4687770` and `rs1404862` involved two causal variants. Inspecting Top PICS posterior probabilities for these variants, we observed a modest spread of values (all contained in [0.33, 0.85]). Each variant was also contained in at least three slices in Stable PICS (three were contained in five slices), while posterior probability support sizes ranged from 6 to 57. Apart from `rs4687770`, which had a considerable difference in allele frequency between YRI and GBR ($f_{YRI} - f_{GBR} = 0.36$), all other variants did not report substantial difference in allele frequencies by ancestry. None of these variants were causal variants used in simulations.

While we cannot derive any generalizable conclusions from such a small number of observations, this analysis suggests that matching variants with very low stable posterior probability are largely depleted in causal variants. However, at least for Potential Set 1, other factors, such as the number of slices including the stable variant as well as the top variant posterior probability, may still be useful for causal variant enrichment.

#### GEUVADIS

Again using a very low stable posterior probability cutoff of 0.01, we found 17 matching variants with very low stable posterior probability across all three potential sets. These are listed in *Appendix 12—table 5*. Briefly, the Top PICS posterior probabilities ranged from 0.23 to 1, with Stable PICS support sizes ranging from as small as 4 to as large as 45. The number of slices containing the stable variant (including the ALL slice) ranged from 3 to 6, while the maximum allele frequency difference between any pair of ancestry slices ranged from 0.084 to 0.43. Comparing these quantities potential set by potential set, allele frequencies tend to be less homogeneous and support sizes tend to be in the upper half of the distribution across the complementary sets of matching variants with stable posterior probability ≥0.01. Notably, all eight Potential Set 1 variants had support sizes lying above the third quartile of the distribution of support sizes for Potential Set 1 matching variants with stable posterior probability ≥0.01. While we cannot derive any generalizable conclusions from such a small number of observations, this suggests that a larger support size could result in stable variants with low stable posterior probability.

Because we do not know the causal variants, we performed functional enrichment analyses of the 17 matching variants. Similar to the above analysis, we cannot make generalizable claims owing to small sample sizes, but we can nonetheless report some patterns. By inspecting each functional annotation, we did not notice systematic patterns in either extreme direction relative to matching variants with stable posterior probability ≥0.01. However, we occasionally found variants that stand out for having impactful functional annotation scores. We list one below for each potential set.

- Potential Set 1 reported the variant `rs12224894` from fine-mapping `ENSG00000255284.1` (accession code *AP006621.3*) in Chromosome 11. This variant stood out for lying in the promoter flanking region of multiple cell types and being relatively enriched for GC content within a 75 bp flanking region. This variant has been reported as a cis eQTL for *AP006621* (using whole blood gene expression, rather than lymphoblastoid cell line gene expression in this study) in a clinical trial study of patients with systemic lupus erythematosus (*Davenport et al., 2018*). Its nearest gene is *GATD1*, a ubiquitously expressed gene that codes for a protein and is predicted to regulate enzymatic and catabolic activity. This variant appeared in all 6 slices, with a moderate support size of 23.

- Potential Set 2 reported the variant `rs9912201` from fine-mapping `ENSG00000108592.9` (mapped to *FTSJ3*) in Chromosome 17. Its FIRE score is 0.976, which is close to the maximum FIRE score reported across all Potential Set 2 matching variants. This variant has been reported as an SNP in high LD to a GWAS hit SNP `rs7223966` in a pan-cancer study (*Gong et al., 2018*). This variant appeared in all six slices, with a moderate support size of 32.

- Potential Set 3 reported the variant `rs625750` from fine-mapping `ENSG00000254614.1` (mapped to *CAPN1-AS1*, an RNA gene) in Chromosome 11. Its FIRE score is 0.971 and its B statistic is 0.405 (region under selection), which lie at the extreme quantiles of the distributions of these scores for Potential Set 3 matching variants with stable posterior probability ≥0.01. Its associated mutation has been predicted to affect transcription factor binding, as computed using several position weight matrices (*Kheradpour and Kellis, 2014*). This variant appeared in just three slices, possibly owing to the considerable allele frequency difference between ancestries (maximum AF difference = 0.22). However, it has a small support size of 4, with moderately high Top PICS posterior probability of 0.64.

To summarize, our analysis of GEUVADIS fine-mapped variants demonstrates that matching variants with very low stable posterior probability could still be functionally important, even for lower potential sets, conditional on supportive scores in interpretable features such as the number of slices containing the stable variant and the posterior probability support size. It would be interesting to scale up such analyses in future work to investigate the generalizability of these conclusions beyond just 17 data points.

## Appendix 12

## Supplementary tables

**Appendix 12—table 1.** Plain and Stable PICS matching frequencies.
Below reports the frequencies with which Plain and Stable PICS have matching variants for the same potential set. The numbers of matching variants for each SNR scenario are reported in the parentheses. The bottom two rows show matching frequencies when results are stratified by posterior probability (PP) of the Plain PICS variant. The numbers of matching variants for each PP stratum are reported in the parentheses.

**Stratified by signal-to-noise ratio (SNR) of simulations**

|  | Potential Set 1 | Potential Set 2 | Potential Set 3 |
|---|---|---|---|
| SNR = 0.053 | 0.736 (265) | 0.803 (289) | 0.797 (287) |
| SNR = 0.111 | 0.775 (279) | 0.753 (271) | 0.758 (273) |
| SNR = 0.25 | 0.903 (325) | 0.714 (257) | 0.728 (262) |
| SNR = 0.667 | 0.906 (326) | 0.753 (271) | 0.744 (268) |

**Stratified by posterior probability (PP) of plain PICS variant**

|  | Potential Set 1 | Potential Set 2 | Potential Set 3 |
|---|---|---|---|
| p > 0.9 | 0.978 (441) | 0.899 (286) | 0.927 (307) |
| p ≤ 0.9 | 0.762 (754) | 0.715 (802) | 0.706 (783) |

**Appendix 12—table 2.** Stable and Top PICS matching frequencies.
Below reports the frequencies with which Stable and Top PICS have matching variants for the same potential set. The numbers of matching variants for each SNR/'No. Causal Variants' scenario are reported in the parentheses.

**Stratified by signal-to-noise ratio (SNR) and No. Causal Variants ($S$) in simulations**

|  |  | Potential Set 1 | Potential Set 2 | Potential Set 3 |
|---|---|---|---|---|
| One causal variant ($S = 1$) | SNR = 0.053 | 0.695 (139) | 0.41 (82) | 0.22 (44) |
|  | SNR = 0.111 | 0.75 (150) | 0.405 (81) | 0.24 (48) |
|  | SNR = 0.25 | 0.825 (165) | 0.45 (90) | 0.26 (52) |
|  | SNR = 0.667 | 0.895 (179) | 0.405 (81) | 0.275 (55) |
| Two causal variants ($S = 2$) | SNR = 0.053 | 0.545 (109) | 0.36 (72) | 0.225 (45) |
|  | SNR = 0.111 | 0.68 (136) | 0.38 (76) | 0.215 (43) |
|  | SNR = 0.25 | 0.79 (158) | 0.435 (87) | 0.27 (54) |
|  | SNR = 0.667 | 0.78 (156) | 0.41 (82) | 0.26 (52) |
| Three causal variants ($S = 3$) | SNR = 0.053 | 0.565 (113) | 0.37 (74) | 0.245 (49) |
|  | SNR = 0.111 | 0.655 (131) | 0.36 (72) | 0.265 (53) |
|  | SNR = 0.25 | 0.72 (144) | 0.39 (78) | 0.22 (44) |
|  | SNR = 0.667 | 0.785 (157) | 0.48 (96) | 0.255 (51) |

**Appendix 12—table 3.** Stable and Top SuSiE matching frequencies.
Below reports the frequencies with which Stable and Top SuSiE have matching variants for the same potential set. The numbers of matching variants for each SNR/'No. Causal Variants' scenario are reported in the parentheses.

**Stratified by signal-to-noise ratio (SNR) and No. Causal Variants ($S$) in simulations**

|  | Potential Set 1 | Potential Set 2 | Potential Set 3 |
|---|---|---|---|

*Appendix 12—table 3 Continued on next page*

*Appendix 12—table 3 Continued*

**Stratified by signal-to-noise ratio (SNR) and No. Causal Variants (S) in simulations**

| | | | | |
|---|---|---|---|---|
| One causal variant (S = 1) | SNR = 0.053 | 0.55 (110) | 0.115 (23) | 0.095 (19) |
| | SNR = 0.111 | 0.725 (145) | 0.14 (28) | 0.095 (19) |
| | SNR = 0.25 | 0.84 (168) | 0.145 (29) | 0.17 (34) |
| | SNR = 0.667 | 0.875 (175) | 0.14 (28) | 0.19 (38) |
| Two causal variants (S = 2) | SNR = 0.053 | 0.505 (101) | 0.095 (19) | 0.065 (13) |
| | SNR = 0.111 | 0.73 (146) | 0.14 (28) | 0.095 (19) |
| | SNR = 0.25 | 0.875 (175) | 0.33 (66) | 0.105 (21) |
| | SNR = 0.667 | 0.875 (175) | 0.425 (85) | 0.115 (23) |
| Three causal variants (S = 3) | SNR = 0.053 | 0.345 (69) | 0.075 (15) | 0.085 (17) |
| | SNR = 0.111 | 0.68 (136) | 0.23 (46) | 0.145 (29) |
| | SNR = 0.25 | 0.795 (159) | 0.37 (74) | 0.185 (37) |
| | SNR = 0.667 | 0.85 (170) | 0.585 (117) | 0.25 (50) |

**Appendix 12—table 4.** Off-diagonal matching frequencies and causal variant recovery.
Below reports the number of Stable and Top PICS non-matching variants that match across different, or 'off-diagonal', potential sets. Frequencies are computed across simulations with the same number of causal variants (S = 1, 2, or 3), with numbers along the yellow-shaded diagonal reporting the number of non-matching variants between the same potential sets. Each off-diagonal element reports both the number of matching variants for the pair of potential sets listed as well as the percentage of these matches that also correspond to the causal variant.

**Simulations with one causal variant**

| | | Top PICS potential set compared against | | |
|---|---|---|---|---|
| | | Potential Set 1 | Potential Set 2 | Potential Set 3 |
| Stable PICS potential set | Potential Set 1 | 167 | 5 (60%) | 3 (67%) |
| | Potential Set 2 | 4 (25%) | 466 | 104 (0.96%) |
| | Potential Set 3 | 0 | 94 (0%) | 601 |

**Simulations with two causal variants**

| | | Top PICS potential set compared against | | |
|---|---|---|---|---|
| | | Potential Set 1 | Potential Set 2 | Potential Set 3 |
| Stable PICS potential set | Potential Set 1 | 241 | 24 (46%) | 5 (60%) |
| | Potential Set 2 | 29 (52%) | 483 | 88 (10%) |
| | Potential Set 3 | 9 (11%) | 84 (13%) | 606 |

**Simulations with three causal variants**

| | | Top PICS potential set compared against | | |
|---|---|---|---|---|
| | | Potential Set 1 | Potential Set 2 | Potential Set 3 |
| Stable PICS potential set | Potential Set 1 | 255 | 29 (66%) | 7 (14%) |
| | Potential Set 2 | 30 (40%) | 480 | 79 (18%) |
| | Potential Set 3 | 4 (0%) | 65 (12%) | 603 |

**Appendix 12—table 5.** List of matching variants with low stable posterior probability.
Below summarizes the genes and potential sets for which Stable and Top PICS returned matching variants, along with SNP-level and fine-mapping features for interpretation. Five statistics are reported: posterior probability of the stable variant (Stable PP); posterior probability of the top

variant (Top PP); posterior probability support size, defined as the number variants with positive probability (Support Size); the number of ancestry slices, including the ALL slice, for which the stable variant had positive posterior probability from running Stable PICS (Number of Slices); the maximum difference in allele frequency between any pair of subpopulations among YRI, TSI, GBR, FIN, and CEU (Max AF Difference).

| Potential Set | Gene | Matching variant | Stable PP | Top PP | Support size | Number of slices | Max AF Difference |
|---|---|---|---|---|---|---|---|
| 1 | ENSG00000134762.11 | rs61731921 | 0.0028 | 0.76 | 23 | 4 | 0.22 |
| 1 | ENSG00000197847.8 | rs7130955 | 0.0075 | 0.23 | 45 | 3 | 0.18 |
| 1 | ENSG00000255284.1 | rs12224894 | 0.0067 | 0.65 | 23 | 6 | 0.14 |
| 1 | ENSG00000104442.5 | rs6995242 | 0.0092 | 0.31 | 42 | 4 | 0.34 |
| 1 | ENSG00000146733.9 | rs10239528 | 0.0031 | 0.53 | 27 | 5 | 0.24 |
| 1 | ENSG00000248468.1 | rs9853505 | 0.0099 | 0.29 | 39 | 3 | 0.43 |
| 1 | ENSG00000122224.10 | rs57449 | 0.0089 | 0.50 | 25 | 4 | 0.31 |
| 1 | ENSG00000134262.8 | rs17464525 | 0.0030 | 0.45 | 23 | 4 | 0.15 |
| 2 | ENSG00000216522.3 | rs5751902 | 0.0052 | 1 | 7 | 3 | 0.16 |
| 2 | ENSG00000108592.9 | rs9912201 | 0.0022 | 0.32 | 32 | 6 | 0.27 |
| 2 | ENSG00000134551.7 | rs7315843 | 0.0019 | 0.58 | 10 | 5 | 0.22 |
| 2 | ENSG00000221947.3 | rs3103860 | 0.0018 | 0.99 | 4 | 4 | 0.084 |
| 2 | ENSG00000081791.4 | rs2270113 | 0.0059 | 0.77 | 11 | 4 | 0.33 |
| 3 | ENSG00000140368.8 | rs62027296 | 0.0069 | 0.29 | 21 | 4 | 0.15 |
| 3 | ENSG00000254614.1 | rs625750 | 0.0017 | 0.64 | 4 | 3 | 0.22 |
| 3 | ENSG00000133835.9 | rs2451818 | 0.0036 | 0.40 | 32 | 3 | 0.41 |
| 3 | ENSG00000158234.8 | rs693293 | 0.0069 | 0.55 | 24 | 4 | 0.13 |

**Appendix 12—table 6.** List of variant annotations with interpretations.

| Functional annotation | Interpretation |
|---|---|
| Distance to Canonical Transcription Start Site (TSS) | - |
| Percent CpG in 75 bp window centered on variant position | - |
| Percent GC in 75 bp window centered on variant position | - |
| CTCF Binding Enrichment | Whether the variant lies within a CTCF binding site region as predicted by Ensembl |
| Enhancer Enrichment | Whether the variant lies within an enhancer region as predicted by Ensembl |
| Open Chromatin Enrichment | Whether the variant lies within an open chromatin region as predicted by Ensembl |
| Promoter Enrichment | Whether the variant lies within a promoter region as predicted by Ensembl |
| TF Binding Enrichment | Whether the variant lies within a TF-binding site region as predicted by Ensembl |
| Promoter Flanking Enrichment | Whether the variant lies within a promoter flanking region as predicted by Ensembl |
| CADD (2 scores) | Whether the variant is likely to be simulated or not, and hence likely deleterious or not. One score is raw while the other is rank-normalized |

*Appendix 12—table 6 Continued on next page*

*Appendix 12—table 6 Continued*

| Functional annotation | Interpretation |
| --- | --- |
| SIFTVal | Whether the variant affects protein function, and hence deleterious |
| Polyphen2 | Posterior probability that the variant is damaging |
| LINSIGHT | Probability that the variant site is under selection, thus having functional consequence |
| PhyloP (3 scores) | Substitution rates measuring cross-species evolutionary conservation at the site of the variant. Each score is computed with respect to a clade (vertebrate, mammal, primate) |
| GerpN | Estimated neutral substitution rate at variant position, with higher value implying greater conservation |
| GerpS | Estimated rejected substitution rate at variant position, with positive value implying a deficit in substitutions |
| B Statistic | Background selection at variant position, with smaller value indicating larger impact of selection |
| FATHMM-XF | Integrative score measuring deleteriousness of the variant |
| Funseq2 | Integrative score measuring deleteriousness of the variant |
| ALoft | Integrative score measuring loss of function associated with the variant |
| FIRE | Integrative score measuring deleteriousness of the variant |
| Magnitude of Effect on Enformer Track Prediction (177 tracks) | Change in prediction of a gene regulatory track when performing in silico mutagenesis on the variant in a 196,608 bp sequence |

