## [Editor Report · eLife Assessment]

This **important** study presents a methodologically rigorous framework for stability-guided fine-mapping, extending PICS and generalizing to methods such as SuSiE, supported by comprehensive simulations and functional enrichment analyses. The evidence is now **convincing**, demonstrating improved causal variant recovery and offering a robust alternative for cross-population fine-mapping. The approach will be of particular interest to statistical geneticists, computational biologists, and biomedical researchers who rely on fine-mapping to interpret genetic association signals.

---

## [Referee Report · Reviewer #1 (Public review)]

Aw et al. have proposed that utilizing stability analysis can be useful for fine-mapping of cross populations. In addition, the authors have performed extensive analyses to understand the cases where the top eQTL and stable eQTL are the same or different via functional data.

Comments on revisions:

The authors have answered all my concerns.

---

## [Referee Report · Reviewer #2 (Public review)]

Aw et al presents a new stability-guided fine-mapping method by extending the previously proposed PICS method. They applied their stability-based method to fine-map cis-eQTLs in the GEUVADIS dataset and compared it against residualization-based approaches. They evaluated the performance of the proposed method using publicly available functional annotations and demonstrated that the variants identified by their stability-based method show enrichment for these functional annotations.

The authors have substantially strengthened the manuscript by addressing the major concerns raised in the initial review. I acknowledge that they have conducted comprehensive simulation studies to show the performance of their proposed approach and that they have extended their approach to SuSiE ("Stable SuSiE") to demonstrate the broader applicability of the stability-guided principle beyond PICS.

One remaining question is the interpretation of matching variants with very low stable posterior probabilities (~0), which the authors have analyzed in detail but without fully conclusive findings. I agree with the authors that this event is relatively rare and the current sample size is limited but this might be something to keep in mind for future studies.

---

## [Author Response]

The following is the authors’ response to the latest reviews:

"One remaining question is the interpretation of matching variants with very low stable posterior probabilities (~0), which the authors have analyzed in detail but without fully conclusive findings. I agree with the authors that this event is relatively rare and the current sample size is limited but this might be something to keep in mind for future studies."

**Fine-mapping stability** – **on matching variants with very low stable posterior probability**

We thank Reviewer 2 for encouraging us to think more about how low stable posterior probability matching variants can be interpreted. We describe a few plausible interpretations, even though – as Reviewer 2 and we have both acknowledged – our present experiments do not point to a clear and conclusive account.

One explanation is that the locus captured by the variant might not be well-resolved, in the sense that many correlated variants exist around the locus. Thus, the variant itself is unlikely causal, but the set of variants in high LD with it may contain the true causal variant, or it's possible that the causal variant itself was not sequenced but lies in that locus. A comparison of LD patterns across ancestries at the locus would be helpful here.

Another explanation rests on the following observation. For a variant to be matching between top and stable PICS and to also have very small stable PP, it has to have the largest PP *after residualization* on the ALL slice but also have positive PP with gene expression on many other slices. In other words, failing to control for potential confounders shrinks the PP. If one assumes that the matching variant is truly causal, then our observation points to an example of negative confounding (aka *suppressor effect*). This can occur when the confounders (PCs) are correlated with allele dosage at the causal variant in a different direction than their correlation with gene expression, so that the crude association between unresidualized gene expression and causal variant allele dosage is biased toward 0.

Although our present study does not allow us to systematically confirm either interpretation – since we found that matching variants were depleted in causal variants in our simulations, violating the second argument, but we also found functional enrichment in analyses of GEUVADIS data though only 17 matching variants with low stable PP were reported – we believe a larger-scale study using larger cohort sizes (at least 1000 individuals per ancestry) and many more simulations (to increase yield of such cases) would be insightful.

———

The following is the authors’ response to the original reviews:

**Reviewer #1:**
Major comments:(1) It would be interesting to see how much fine-mapping stability can improve the fine-mapping results in cross-population. One can simulate data using true genotype data and quantify the amount the fine-mapping methods improve utilizing the stability idea.

We agree, and have performed simulation studies where we assume that causal variants are shared across populations. Specifically, by mirroring the simulation approach described in Wang et al. (2020), we generated 2,400 synthetic gene expression phenotypes across 22 autosomes, using GEUVADIS gene expression metadata (i.e., gene transcription start site) to ensure largely *cis* expression phenotypes were simulated. We additionally generated 1,440 synthetic gene expression phenotypes that incorporate environmental heterogeneity, to motivate our pursuit of fine-mapping stability in the first place (see Response to Reviewer 2, Comment 6). These are described in Results section “Simulation study”:

We evaluated the performance of the PICS algorithm, specifically comparing the approach incorporating stability guidance against the residualization approach that is more commonly used — similar to our application to the real GEUVADIS data. We additionally investigated two ways of “combining” the residualization and stability guidance approaches: (1) running stability-guided PICS on residualized phenotypes; (2) prioritizing matching variants returned by both approaches. See Response to Reviewer 2, Comment 5.

(2) I would be very interested to see how other fine-mapping methods (FINEMAP, SuSiE, and CAVIAR) perform via the stability idea.

Thank you for this valuable comment. We ran SuSiE on the same set of simulated datasets. Specifically, we ran a version that uses residualized phenotypes (supposedly removing the effects of population structure), and also a version that incorporates stability. The second version is similar to how we incorporate stability in PICS. We investigated the performance of Stable SuSiE in a similar manner to our investigation of PICS. First we compared the performance relative to SuSiE that was run on residualized phenotypes. Motivated by our finding in PICS that prioritizing matching variants improves causal variant recovery, we did the same analysis for SuSiE. This analysis is described in Results section “Stability guidance improves causal variant recovery in SuSiE.”

We reported overall matching frequencies and causal variant recovery rates of top and stable variants for SuSiE in Figures 2C&D.

Frequencies with which Stable and Top SuSiE variants match, stratified by the simulation parameters, are summarized in Supplementary File 2C (reproduced for convenience in Response to Reviewer 2, Comment 3). Causal variant recovery rates split by the number of causal variants simulated, and stratified by both signal-to-noise ratio and the number of credible sets included, are reported in Figure 2—figure supplements 16-18. We reproduce Figure 2—figure supplement 18 (three causal variants scenario) below for convenience. Analogous recovery rates for matching versus non-matching top or stable variants are reported in Figure 2—figure supplements 19, 21 and 23.

(3) I am a little bit concerned about the PICS's assumption about one causal variant. The authors mentioned this assumption as one of their method limitations. However, given the utility of existing fine-mapping methods (FINEMAP and SuSiE), it is worth exploring this domain.

Thank you for raising this fair concern. We explored this domain, by considering simulations that include two and three causal variants (see Response to Reviewer 2, Comment 3). We looked at how well PICS recovers causal variants, and found that each potential set largely does not contain more than one causal variant (Figure 2—figure supplements 20 and 22). This can be explained by the fact that PICS potential sets are constructed from variants with a minimum linkage disequilibrium to a focal variant. On the other hand, in SuSiE, we observed multiple causal variants appearing in lower credible sets when applying stability guidance (Figure 2—figure supplements 21 and 23). A more extensive study involving more fine-mapping methods and metrics specific to violation of the one causal variant assumption could be pursued in future work.

**Reviewer #2:**
Aw et al. presents a new stability-guided fine-mapping method by extending the previously proposed PICS method. They applied their stability-based method to fine-map cis-eQTLs in the GEUVADIS dataset and compared it against what they call residualization-based method. They evaluated the performance of the proposed method using publicly available functional annotations and claimed the variants identified by their proposed stability-based method are more enriched for these functional annotations.While the reviewer acknowledges the contribution of the present work, there are a couple of major concerns as described below.Major:(1) It is critical to evaluate the proposed method in simulation settings, where we know which variants are truly causal. While I acknowledge their empirical approach using the functional annotations, a more unbiased, comprehensive evaluation in simulations would be necessary to assess its performance against the existing methods.

Thank you for this point. We agree. We have performed a simulation study where we assume that causal variants are shared across populations (see response to Reviewer 1, Comment 1). Specifically, by mirroring the simulation approach described in Wang et al. (2020), we generated 2,400 synthetic gene expression phenotypes across 22 autosomes, using GEUVADIS gene expression metadata (i.e., gene transcription start site) to ensure *cis* expression phenotypes were simulated.

(2) Also, simulations would be required to assess how the method is sensitive to different parameters, e.g., LD threshold, resampling number, or number of potential sets.

Thank you for raising this point. The underlying PICS algorithm was not proposed by us, so we followed the default parameters set (LD threshold, r^2^ = 0.5; see Taylor et al., 2021 Bioinformatics) to focus on how stability considerations will impact the existing fine-mapping algorithm. We attempted to derive the asymptotic joint distribution of the p-values, but it was too difficult. Hence, we used 500 permutations because such a large number would allow large-sample asymptotics to kick in. However, following your critical suggestion we varied the number of potential sets in our analyses of simulated data. We briefly mention this in the Results.

“In the Supplement, we also describe findings from investigations into the impact of including more potential sets on matching frequency and causal variant recovery…”

A detailed write-up is provided in Supplementary File 1 Section S2 (p.2):

“The number of credible or potential sets is a parameter in many fine-mapping algorithms. Focusing on stability-guided approaches, we consider how including more potential sets for stable fine-mapping algorithms affects both causal variant recovery and matching frequency in simulations…

Causal variant recovery. We investigate both Stable PICS and Stable SuSiE. Focusing first on simulations with one causal variant, we observe a modest gain in causal variant recovery for both Stable PICS and Stable SuSiE, most noticeably when the number of sets was increased from 1 to 2 under the lowest signal-to-noise ratio setting…”

We observed that increasing the number of potential sets helps with recovering causal variants for Stable PICS (Figure 2—figure supplements 13-15). This observation also accounts for the comparable power that Stable PICS has with SuSiE in simulations with low signal-to-noise ratio (SNR), when we increase the number of credible sets or potential sets (Figure 2—figure supplements 10-12).

(3) Given the previous studies have identified multiple putative causal variants in both GWAS and eQTL, I think it's better to model multiple causal variants in any modern fine-mapping methods. At least, a simulation to assess its impact would be appreciated.

We agree. In our simulations we considered up to three causal variants in *cis*, and evaluated how well the top three Potential Sets recovered all causal variants (Figure 2—figure supplements 13-15; Figure 2—figure supplement 15). We also reported the frequency of variant matches between Top and Stable PICS stratified by the number of causal variants simulated in Supplementary File 2B and 2C. Note Supplementary File 2C is for results from SuSiE fine-mapping; see Response to Reviewer 1, Comment 2.

Supplementary File 2B. Frequencies with which Stable and Top PICS have matching variants for the same potential set. For each SNR/ “No. Causal Variants” scenario, the number of matching variants is reported in parentheses.

Supplementary File 2C. Frequencies with which Stable and Top SuSiE have matching variants for the same credible set. For each SNR/ “No. Causal Variants” scenario, the number of matching variants is reported in parentheses.

(4) Relatedly, I wonder what fraction of non-matching variants are due to the lack of multiple causal variant modeling.

PICS handles multiple causal variants by including more potential sets to return, owing to the important caveat that causal variants in high LD cannot be statistically distinguished. For example, if one believes there are three causal variants that are not too tightly linked, one could make PICS return three potential sets rather than just one. To answer the question using our simulation study, we subsetted our results to just scenarios where the top and stable variants do not match. This mimics the exact scenario of having modeled multiple causal variants but still not yielding matching variants, so we can investigate whether these non-matching variants are in fact enriched in the true causal variants.

Because we expect causal variants to appear in some potential set, we specifically considered whether these non-matching causal variants might match along different potential sets across the different methods. In other words, we compared the stable variant with the top variant from another potential set for the other approach (e.g., Stable PICS Potential Set 1 variant vs Top PICS Potential Set 2 variant). First, we computed the frequency with which such pairs of variants match. A high frequency would demonstrate that, even if the corresponding potential sets do not have a variant match, there could still be a match between non-corresponding potential sets across the two approaches, which shows that multiple causal variant modeling boosts identification of matching variants between both approaches — regardless of whether the matching variant is in fact causal.

Low frequencies were observed. For example, when restricting to simulations where Top and Stable PICS Potential Set 1 variants did not match, about 2-3% of variants matched between the Potential Set 1 variant in Stable PICS and Potential Sets 2 and 3 variants in Top PICS; or between the Potential Set 1 variant in Top PICS and Potential Sets 2 and 3 variants in Stable PICS (Supplementary File 2D). When looking at non-matching Potential Set 2 or Potential Set 3 variants, we do see an increase in matching frequencies (between 10-20%) between Potential Set 2 variants and other potential set variants between the different approaches. However, these percentages are still small compared to the matching frequencies we observed between corresponding potential sets (e.g., for simulations with one causal variant this was 70-90% between Top and Stable PICS Potential Set 1, and for simulations with two and three causal variants this was 55-78% and 57-79% respectively).

We next checked whether these “off-diagonal” matching variants corresponded to the true causal variants simulated. Here we find that the causal variant recovery rate is mostly less than the corresponding rate for diagonally matching variants, which together with the low matching frequency suggests that the enrichment of causal variants of “off-diagonal” matching variants is much weaker than in the diagonally matching approach. In other words, the fraction of non-matching (causal) variants due to the lack of multiple causal variant modeling is low.

We discuss these findings in Supplementary File 1 Section S2 (bottom of p.2).

(5) I wonder if you can combine the stability-based and the residualization-based approach, i.e., using the residualized phenotypes for the stability-based approach. Would that further improve the accuracy or not?

This is a good idea, thank you for suggesting it. We pursued this combined approach on simulated gene expression phenotypes, but did not observe significant gains in causal variant recovery (Figure 2B; Figure 2—figure supplements 2, 13 and 15). We reported this Results “Searching for matching variants between Top PICS and Stable PICS improves causal variant Recovery.”

“We thus explore ways to combine the residualization and stability-driven approaches, by considering (i) combining them into a single fine-mapping algorithm (we call the resulting procedure Combined PICS); and (ii) prioritizing matching variants between the two algorithms. Comparing the performance of Combined PICS against both Top and Stable PICS, however, we find no significant difference in its ability to recover causal variants (Figure 2B)...”

However, we also confirmed in our simulations that prioritizing matching variants between the two approaches led to gains in causal variant recovery (Figure 2D; Figure 2—figure supplements 4, 19, 20 and 22). We reported this Results “Searching for matching variants between Top PICS and Stable PICS improves causal variant Recovery.”

“On the other hand, matching variants between Top and Stable PICS are significantly more likely to be causal. Across all simulations, a matching variant in Potential Set 1 is 2.5X as likely to be causal than either a non-matching top or stable variant (Figure 2D) — a result that was qualitatively consistent even when we stratified simulations by SNR and number of causal variants simulated (Figure 2—figure supplements 19, 20 and 22)...”

This finding is consistent with our analysis of real GEUVADIS gene expression data, where we reported larger functional significance of matching variants relative to non-matching variants returned by either Top of Stable PICS.

(6) The authors state that confounding in cohorts with diverse ancestries poses potential difficulties in identifying the correct causal variants. However, I don't see that they directly address whether the stability approach is mitigating this. It is hard to say whether the stability approach is helping beyond what simpler post-hoc QC (e.g., thresholding) can do.

Thank you for raising this fair point. Here is a model we have in mind. Gene expression phenotypes (Y) can be explained by both genotypic effects (G, as in genotypic allelic dosage) and the environment (E): Y = G + E. However, both G and E depend on ancestry (A), so that Y = G|A+E|A. Suppose that the causal variants are shared across ancestries, so that (G|A=*a*)=G for all ancestries *a*. Suppose however that environments are heterogeneous by ancestry: (E|A=*a*) = e(*a*) for some function e that depends non-trivially on *a*. This would violate the exchangeability of exogenous E in the full sample, but by performing fine-mapping on each ancestry stratum, the exchangeability of exogenous E is preserved. This provides theoretical justification for the stability approach.

We next turned to simulations, where we investigated 1,440 simulated gene expression phenotypes capturing various ways in which ancestry induces heterogeneity in the exogenous E variable (simulation details in Lines 576-610 of Materials and Methods). We ran Stable PICS, as well as a version of PICS that did not residualize phenotypes or apply the stability principle. We observed that (i) causal variant recovery performance was not significantly different between the two approaches (Figure 2—figure supplements 24-32); but (ii) disagreement between the approaches can be considerable, especially when the signal-to-noise ratio is low (Supplementary File 2A). For example, in a set of simulations with three causal variants, with SNR = 0.11 and E heterogeneous by ancestry by letting E be drawn from *N*(2σ,σ^2^) for only GBR individuals (rest are *N*(0,σ^2^)), there was disagreement between Potential Set 1 and 2 variants in 25% of simulations — though recovery rates were similar (Probability of recovering at least one causal variant: 75% for Plain PICS and 80% for Stable PICS). These points suggest that confounding in cohorts can reduce power in methods not adjusting or accounting for ancestral heterogeneity, but can be remedied by approaches that do so. We report this analysis in Results “Simulations justify exploration of stability guidance”

In the current version of our work, we have evaluated, using both simulations and empirical evidence, different ways to combine approaches to boost causal variant recovery. Our simulation study shows that prioritizing matching variants across multiple methods improves causal variant recovery. On GEUVADIS data, where we might not know which variants are causal, we already demonstrated that matching variants are enriched for functional annotations. Therefore, our analyses justify that the adverse consequence of confounding on reducing fine-mapping accuracy can be mitigated by prioritizing matching variants between algorithms including those that account for stability.

(7) For non-matching variants, I wonder what the difference of posterior probabilities is between the stable and top variants in each method. If the difference is small, maybe it is due to noise rather than signal.

We have reported differences in posterior probabilities returned by Stable and Top PICS for GEUVADIS data; see Figure 3—figure supplement 1. For completeness, we compute the differences in posterior probabilities and summarize these differences both as histograms and as numerical summary statistics.

Potential Set 1

- Number of non-matching variants = 9,921

- Table of Summary Statistics of (Stable Posterior Probability – Top Posterior Probability)

**Author response table 1. sa3table1:** 

Min	1st Qu.	Median	Mean	3rd Qu.	Max
-0.999	-0.342	-0.107	-0.117	0.067	0.949

- Histogram of (Stable Posterior Probability – Top Posterior Probability)

Potential Set 2

- Number of non-matching variants = 14,454

- Table of Summary Statistics of (Stable Posterior Probability – Top Posterior Probability)

**Author response table 2. sa3table2:** 

Min	1st Qu.	Median	Mean	3rd Qu.	Max
-1.000	-0.334	-0.0741	-0.0778	0.162	0.976

- Histogram of (Stable Posterior Probability – Top Posterior Probability)

**Author response image 2. sa3fig2:** 

Potential Set 3

- Number of non-matching variants = 16,814

- Table of Summary Statistics of (Stable Posterior Probability – Top Posterior Probability)

**Author response table 3. sa3table3:** 

Min	1st Qu.	Median	Mean	3rd Qu.	Max
-0.998	-0.327	-0.0564	-0.0629	0.191	0.968

- Histogram of (Stable Posterior Probability – Top Posterior Probability)

**Author response image 3. sa3fig3:** 

We also compared the difference in posterior probabilities between non-matching variants returned by Stable PICS and Top PICS for our 2,400 simulated gene expression phenotypes. Focusing on just Potential Set 1 variants, we find two equally likely scenarios, as demonstrated by two distinct clusters of points in a “posterior probability-posterior probability” plot. The first is, as pointed out, a small difference in posterior probability (points lying close to y=x). The second, however, reveals stable variants with very small posterior probability (of order 4 x 10^–5^ to 0.05) but with a non-matching top variant taking on posterior probability well distributed along [0,1]. Moving down to Potential Sets 2 and 3, the distribution of pairs of posterior probabilities appears less clustered, indicating less tendency for posterior probability differences to be small (Figure 2—figure supplement 8).

Here are the histograms and numerical summary statistics.

Potential Set 1

- Number of non-matching variants = 663 (out of 2,400)

- Table of Summary Statistics of (Stable Posterior Probability – Top Posterior Probability)

**Author response table 4. sa3table4:** 

Min	1st Qu.	Median	Mean	3rd Qu.	Max
-0.994	-0.266	-0.00645	-0.109	0.049	0.924

- Histogram of (Stable Posterior Probability – Top Posterior Probability)

**Author response image 4. sa3fig4:** 

Potential Set 2

Number of non-matching variants = 1,429 (out of 2,400)

- Table of Summary Statistics of (Stable Posterior Probability – Top Posterior Probability)

**Author response table 5. sa3table5:** 

Min	1st Qu.	Median	Mean	3rd Qu.	Max
-0.998	-0.376	-0.0944	-0.121	0.115	0.903

- Histogram of (Stable Posterior Probability – Top Posterior Probability)

**Author response image 5. sa3fig5:** 

Potential Set 3

- Number of non-matching variants = 1,810 (out of 2,400)

- Table of Summary Statistics of (Stable Posterior Probability – Top Posterior Probability)

**Author response table 6. sa3table6:** 

Min	1st Qu.	Median	Mean	3rd Qu.	Max
-0.998	-0.389	-0.100	-0.114	0.146	0.915

- Histogram of (Stable Posterior Probability – Top Posterior Probability)

**Author response image 6. sa3fig6:** 

(8) It's a bit surprising that you observed matching variants with (stable) posterior probability ~ 0 (SFig. 1). What are the interpretations for these variants? Do you observe functional enrichment even for low posterior probability matching variants?

Thank you for this question. We have performed a thorough analysis of matching variants with very low stable posterior probability, which we define as having a posterior probability < 0.01 (Supplementary File 1 Section S11). Here, we briefly summarize the analysis and key findings.

Analysis

First, such variants occur very rarely — only 8 across all three potential sets in simulations, and 17 across all three potential sets for GEUVADIS (the latter variants are listed in Supplementary 2E). We begin interpreting these variants by looking at allele frequency heterogeneity by ancestry, support size — defined as the number of variants with positive posterior probability in the ALL slice* — and the number of slices including the stable variant (i.e., the stable variant reported positive posterior probability for the slice).

*Note that the stable variant posterior probability need not be at least 1/(Support Size). This is because the algorithm may have picked a SNP that has a lower posterior probability in the ALL slice (i.e., not the top variant) but happens to appear in the most number of other slices (i.e., a stable variant).

For variants arising from simulations, because we know the true causal variants, we check if these variants are causal. For GEUVADIS fine-mapped variants, we rely on functional annotations to compare their relative enrichment against other matching variants that did not have very low stable posterior probability.

Findings

While we caution against generalizing from observations reported here, which are based on very small sample sizes, we noticed the following. In simulations, matching variants with very low stable posterior probability are largely depleted in causal variants, although factors such as the number of slices including the stable variant may still be useful. In GEUVADIS, however, these variants can still be functionally enriched. We reported three examples in Supplementary File 1 Section S11 (pp. 8-9 of Supplement), where the variants were enriched in either VEP or biologically interpretable functional annotations, and were also reported in earlier studies. We partially reproduce our report below for convenience.

“However, we occasionally found variants that stand out for having large functional annotation scores. We list one below for each potential set.

- Potential Set 1 reported the variant rs12224894 from fine-mapping ENSG00000255284.1 (accession code *AP006621.3*) in Chromosome 11. This variant stood out for lying in the promoter flanking region of multiple cell types and being relatively enriched for GC content with a 75bp flanking region. This variant has been reported as a cis eQTL for AP006632 (using whole blood gene expression, rather than lymphoblastoid cell line gene expression in this study) in a clinical trial study of patients with systemic lupus erythematosus (Davenport et al., 2018). Its nearest gene is GATD1, a ubiquitously expressed gene that codes for a protein and is predicted to regulate enzymatic and catabolic activity. This variant appeared in all 6 slices, with a moderate support size of 23.

- Potential Set 2 reported the variant rs9912201 from fine-mapping ENSG00000108592.9 (mapped to *FTSJ3*) in Chromosome 17. Its FIRE score is 0.976, which is close to the maximum FIRE score reported across all Potential Set 2 matching variants. This variant has been reported as a SNP in high LD to a GWAS hit SNP rs7223966 in a pan-cancer study (Gong et al., 2018). This variant appeared in all 6 slices, with a moderate support size of 32.

- Potential Set 3 reported the variant rs625750 from fine-mapping ENSG00000254614.1 (mapped to *CAPN1-AS1*, an RNA gene) in Chromosome 11. Its FIRE score is 0.971 and its B statistic is 0.405 (region under selection), which lie at the extreme quantiles of the distributions of these scores for Potential Set 3 matching variants with stable posterior probability at least 0.01. Its associated mutation has been predicted to affect transcription factor binding, as computed using several position weight matrices (Kheradpour and Kellis, 2014). This variant appeared in just 3 slices, possibly owing to the considerable allele frequency difference between ancestries (maximum AF difference = 0.22). However, it has a small support size of 4 and a moderately high Top PICS posterior probability of 0.64.

To summarize, our analysis of GEUVADIS fine-mapped variants demonstrates that matching variants with very low stable posterior probability could still be functionally important, even for lower potential sets, conditional on supportive scores in interpretable features such as the number of slices containing the stable variant and the posterior probability support size…”